# Persistent symptoms and clinical findings in adults with post-acute sequelae of COVID-19/post-COVID-19 syndrome in the second year after acute infection: A population-based, nested case-control study

Raphael S. Peter[1‡], Alexandra Nieters[2‡], Siri Göpel[3], Uta Merle[4], Jürgen M. Steinacker[5], Peter Deibert[6], Birgit Friedmann-Bette[7], Andreas Nieß[8], Barbara Müller[9], Claudia Schilling[10], Gunnar Erz[8], Roland Giesen[11], Veronika Götz[11], Karsten Keller[7], Philipp Maier[6], Lynn Matits[5], Sylvia Parthé[9], Martin Rehm[1], Jana Schellenberg[5], Ulrike Schempf[3], Mengyu Zhu[4,12], Hans-Georg Kräusslich[9,13], Dietrich Rothenbacher[1‡], Winfried V. Kern[11‡*], on behalf of the EPILOC Phase 2 Study Group[¶]

1 Institute of Epidemiology and Medical Biometry, Ulm University, Ulm, Germany, 2 Institute for Immunodeficiency, Medical Centre and Faculty of Medicine, Albert-Ludwigs-University, Freiburg, Germany, 3 Department of Internal Medicine I, University Hospital Tübingen, Tübingen, Germany, 4 Department of Internal Medicine IV, Heidelberg University Faculty of Medicine and Heidelberg University Hospital, Heidelberg, Germany, 5 Division of Sports and Rehabilitation Medicine, Department of Medicine, Ulm University Hospital, Ulm, Germany, 6 Institute for Exercise and Occupational Medicine, Medical Centre and Faculty of Medicine, Albert-Ludwigs-University, Freiburg, Germany, 7 Department of Sports Medicine, Heidelberg University Faculty of Medicine and Heidelberg University Hospital, Heidelberg, Germany, 8 Department of Sports Medicine, University Hospital Tübingen, Tübingen, Germany, 9 Department of Infectious Diseases—Virology, Heidelberg University Faculty of Medicine, and Heidelberg University Hospital, Heidelberg, Germany, 10 Department of Psychiatry and Psychotherapy, Sleep Laboratory, Medical Faculty Mannheim, Central Institute of Mental Health (ZI), University of Heidelberg, Heidelberg, Germany, 11 Division of Infectious Diseases, Department of Medicine II, Medical Centre and Faculty of Medicine, Albert-Ludwigs-University, Freiburg, Germany, 12 Institute for Cardiovascular Prevention, Ludwig-Maximilians-University, Munich, Germany, 13 German Centre for Infection Research (DZIF) Partner Site Heidelberg, Heidelberg, Germany

¶Membership of the EPILOC Phase 2 Study Group is listed in the Acknowledgments.
‡ RSP and AN are the co-first authors on this work. DR and WVK are joint senior authors on this work.
* winfried.kern@uniklinik-freiburg.de

## Abstract

### Background

Self-reported health problems following severe acute respiratory syndrome coronavirus 2 (SARS-CoV-2) infection are common and often include relatively non-specific complaints such as fatigue, exertional dyspnoea, concentration or memory disturbance and sleep problems. The long-term prognosis of such post-acute sequelae of COVID-19/post-COVID-19 syndrome (PCS) is unknown, and data finding and correlating organ dysfunction and pathology with self-reported symptoms in patients with non-recovery from PCS is scarce. We wanted to describe clinical characteristics and diagnostic findings among patients with PCS persisting for >1 year and assessed risk factors for PCS persistence versus improvement.

**Data availability statement:** Data can be made available from the University of Ulm (via: duac.EPILOC@uni-ulm.de) for researchers who meet the criteria for access to confidential data.

**Funding:** This work was funded by a Baden-Württemberg Federal State Ministry of Science and Art (https://mwk.baden-wuerttemberg.de) grant (number MR/S028188/1) to WVK, HGK, UM, DR, SG and JS. The funder had no role in study design, data collection and analysis, decision to publish, or preparation of the manuscript.

**Competing interests:** The authors have declared that no competing interests exist.

**Abbreviations:** ACTH, adrenocorticotropic hormone; AUC, area under the curve; BMI, body mass index; BR, breathing reserve; CFQ-11, Chalder Fatigue Scale; CH50, complement haemolytic activity; CI, confidence interval; CMV, cytomegalovirus; COMPASS-31, Composite Autonomic Symptom Score 31; CPET, cardiopulmonary exercise testing; CRP, C-reactive protein; DHEA-S, dehydroepiandrosterone sulfate; EA-D, early antigen D; EBNA, EBV nuclear antigen; EBV, Epstein–Barr virus; ECGs, electrocardiograms; EPILOC, Epidemiology of Long Covid; ESS, Epworth Sleepiness Scale; FEV1, forced expiratory volume in one second; FLei, "Fragebogen zur geistigen Leistungsfähigkeit"; FVC, forced vital capacity; GAD-7, Generalised Anxiety Disorder 7; HbA1c, glycated haemoglobin; HR, heart rate; ISI, Insomnia Severity Index; LV-E/A, ratio of maximal early to late diastolic transmitral flow velocity; LV-E/e', ratio between early mitral inflow and mitral annular early diastolic velocities; LV-EF, left ventricular volume and ejection fraction; ME/CFS, myalgic encephalomyelitis (or encephalopathy)/chronic fatigue syndrome; mMRC, modified-Medical Research Council Dyspnoea Scale; MoCA, Montreal Cognitive Assessment; N, nucleocapsid; PCS, post-COVID-19 syndrome; PEM, post-exertional malaise; PHQ-9, Patient Health Questionnaire 9; pro-BNP, pro-brain natriuretic peptide; PSQI, Pittsburgh Sleep Quality Index; PSS-10, 10-item Perceived Stress Scale; RER, respiratory exchange ratio; RT-PCR, reverse transcription polymerase chain reaction; SARS-CoV-2, severe acute respiratory syndrome coronavirus 2; SDMT, Symbol Digit Modalities Test; SF-12, Short Form-12 Health Survey; $SpO_2$, peripheral oxygen saturation; STROBE, Strengthening the Reporting of Observational Studies in Epidemiology; TMT-B, Trail making test part B; TSH, thyroid-stimulating hormone; VCA, viral capsid antigen; VE/$VCO_2$ slope, slope of minute ventilation to carbon dioxide production; $VO_{2max}$, oxygen uptake.

## Methods and findings

This nested population-based case-control study included subjects with PCS aged 18–65 years with ($n = 982$) and age- and sex-matched control subjects without PCS ($n = 576$) according to an earlier population-based questionnaire study (6–12 months after acute infection, phase 1) consenting to provide follow-up information and to undergo comprehensive outpatient assessment, including neurocognitive, cardiopulmonary exercise, and laboratory testing in four university health centres in southwestern Germany (phase 2, another 8.5 months [median, range 3–14 months] after phase 1). The mean age of the participants was 48 years, and 65% were female. At phase 2, 67.6% of the patients with PCS at phase 1 developed persistent PCS, whereas 78.5% of the recovered participants remained free of health problems related to PCS. Improvement among patients with earlier PCS was associated with mild acute index infection, previous full-time employment, educational status, and no specialist consultation and not attending a rehabilitation programme. The development of new symptoms related to PCS among participants initially recovered was associated with an intercurrent secondary SARS-CoV-2 infection and educational status. Patients with persistent PCS were less frequently never smokers (61.2% versus 75.7%), more often obese (30.2% versus 12.4%) with higher mean values for body mass index (BMI) and body fat, and had lower educational status (university entrance qualification 38.7% versus 61.5%) than participants with continued recovery. Fatigue/exhaustion, neurocognitive disturbance, chest symptoms/breathlessness and anxiety/depression/sleep problems remained the predominant symptom clusters. Exercise intolerance with post-exertional malaise (PEM) for >14 h and symptoms compatible with myalgic encephalomyelitis/chronic fatigue syndrome were reported by 35.6% and 11.6% of participants with persistent PCS patients, respectively. In analyses adjusted for sex-age class combinations, study centre and university entrance qualification, significant differences between participants with persistent PCS versus those with continued recovery were observed for performance in three different neurocognitive tests, scores for perceived stress, subjective cognitive disturbances, dysautonomia, depression and anxiety, sleep quality, fatigue and quality of life. In persistent PCS, handgrip strength (40.2 [95% confidence interval (CI) [39.4, 41.1]] versus 42.5 [95% CI [41.5, 43.6]] kg), maximal oxygen consumption (27.9 [95% CI [27.3, 28.4]] versus 31.0 [95% CI [30.3, 31.6]] ml/min/kg body weight) and ventilatory efficiency (minute ventilation/carbon dioxide production slope, 28.8 [95% CI [28.3, 29.2]] versus 27.1 [95% CI [26.6, 27.7]]) were significantly reduced relative to the control group of participants with continued recovery after adjustment for sex-age class combinations, study centre, education, BMI, smoking status and use of beta blocking agents. There were no differences in measures of systolic and diastolic cardiac function at rest, in the level of N-terminal brain natriuretic peptide blood levels or other laboratory measurements (including complement activity, markers of Epstein–Barr virus [EBV] reactivation, inflammatory and coagulation markers, serum levels of cortisol, adrenocorticotropic hormone and dehydroepiandrosterone sulfate). Screening for viral persistence (PCR in stool samples and SARS-CoV-2 spike antigen levels in plasma) in a subgroup of the patients with persistent PCS was negative. Sensitivity analyses (pre-existing illness/comorbidity, obesity, medical care of the index acute infection) revealed similar findings. Patients with persistent PCS and PEM reported more pain symptoms and had worse results in almost all tests. A limitation was that we had no objective information

on exercise capacity and cognition before acute infection. In addition, we did not include patients unable to attend the outpatient clinic for whatever reason including severe illness, immobility or social deprivation or exclusion.

## Conclusions

In this study, we observed that the majority of working age patients with PCS did not recover in the second year of their illness. Patterns of reported symptoms remained essentially similar, non-specific and dominated by fatigue, exercise intolerance and cognitive complaints. Despite objective signs of cognitive deficits and reduced exercise capacity, there was no major pathology in laboratory investigations, and our findings do not support viral persistence, EBV reactivation, adrenal insufficiency or increased complement turnover as pathophysiologically relevant for persistent PCS. A history of PEM was associated with more severe symptoms and more objective signs of disease and might help stratify cases for disease severity.

## Author summary

### Why was this study done?

- Self-reported health problems following severe acute respiratory syndrome coronavirus 2 (SARS-CoV-2) infection have commonly been described and may persist for months. They typically include relatively non-specific complaints such as fatigue, exertional dyspnoea, concentration or memory disturbance and sleep problems.

- The long-term prognosis of this post-COVID-19 syndrome (PCS) is unknown. To the best of our knowledge, measurable single or multiple organ dysfunction and pathology and their correlation with self-reported symptoms in patients with non-recovery from PCS for more than a year have not been well described.

### What did the researchers do and find?

- We invited individuals who had participated in a previous population-based survey of post-acute complaints and symptoms after acute SARS-CoV-2 infection to undergo a follow-up investigation that included a comprehensive medical evaluation. Results were compared between patients with persistent PCS (cases) and those study participants who had not developed PCS (controls).

- We found that two-thirds of the individuals with PCS had persisting disease for more than a year with no major changes in symptom clusters.

- Objective signs of organ dysfunction and pathology among individuals with persistent PCS correlated with self-reported symptoms, were detected more often among PCS patients with longer lasting post-exertional malaise, and included both reduced physical exercise capacity and reduced cognitive test performances while we did not find differences in the results of multiple laboratory investigations after adjustment for possible confounders such as body mass index and educational status.

- The severity of the index infection, lower educational status, no previous full-time employment and (need for) specialist consultation or a rehabilitation programme (the latter probably due to reverse causation) were factors for non-recovery from PCS.

### What do these findings mean?

- In the majority of patients, PCS symptoms did not improve in the second year of their illness and typically continued to include fatigue and measurable exercise intolerance and cognition deficits, but there seems to be no major pathology in laboratory investigations. Sociodemographic variables appear to play a role not only for the development, but also for the non-recovery from PCS.

- Limitations include the missing information on pathology before acute infection, response and recall biases.

## Introduction

The severe acute respiratory syndrome coronavirus 2 (SARS-CoV-2) pandemic resulted in over 750 million confirmed cases worldwide [1]. Besides morbidity and mortality in the acute phase of the infection, considerable post-acute health problems and sequelae are reported [2–5]. The WHO defined post-coronavirus disease 2019 [COVID-19] condition as the continuation or development of new symptoms after acute SARS-CoV-2 infection, lasting for at least 2 months, and being unexplained by an alternative diagnosis [6]. Slightly different definitions and alternative wording (such as long COVID-19), post-acute sequelae of SARS-CoV-2 infection or post-COVID-19 syndrome [PCS]) have been used [7,8], and are in part relevant for the widely differing prevalence estimates in previous studies [9]. Furthermore, some prevalence estimates may have been biased since many of the early studies focussed on hospitalised or healthcare-seeking patients only [10–12], although most COVID-19 patients do not require medical treatment for the acute infection. Further limitations have been the difficulty of including an uninfected control group to estimate background prevalence of symptoms. In fact, many studies have assessed PCS prevalence and trajectories by using various questionnaires asking for self-reported health problems. Although many of the symptoms may impact everyday functioning, health-related quality of life and work ability [3,4], they lack specificity (i.e., they can have many other causes and overlap with other conditions), are usually not well evaluable in claims data studies and have often not been validated through systematic protocol-prespecified diagnostic studies.

More recently, several diagnostic studies have been able to confirm some impaired neurocognitive functions in patients with PCS [13–17], while the results for cardiac and pulmonary function tests have been variable and less consistent [18,19]. Laboratory studies have suggested a number of altered blood biomarkers (such as various cytokines/chemokines, immune cell markers, plasma metabolites and cortisol) with potential pathophysiologic and diagnostic relevance in PCS patients [20–23]. Many of the clinical or laboratory diagnostic studies; however, were small, lacked appropriate controls, adjustments (e.g., for age and sex, smoking and body composition, educational or socioeconomic status, severity of the acute infection and pre-existing or concomitant disease), or showed only subtle changes compared to controls. Higher body mass index (BMI), for example, has been predictive for persisting dyspnoea in COVID-19 patients [24]. Obesity has been reported as a risk factor for PCS [10,25,26], and mechanistic evidence of why obesity could make people more susceptible to PCS has been provided [27]. Outside the COVID-19 context, BMI in association with sex has been found to be a major confounder in studies of proinflammatory markers [28], and obesity itself has also been associated with cognitive dysfunction [29]. Cognitive dysfunction, interestingly, has been measurable after COVID-19 in individuals who were asymptomatic or had no more symptoms than age- and sex-matched uninfected controls [30,31]. Symptom-based phenotypic stratification of PCS, although attractive and intriguing, thus, may be misleading in

diagnostic studies if not evaluated against adequate controls and if not adjusted for potential confounders.

The aim of the present study was to medically validate PCS among individuals having participated in our previous population-based study of SARS-CoV-2 infected adults (6–12 months after infection) and having been considered to have the syndrome based on self-reported new symptoms with at least moderate impairment in daily life plus either impaired general health or work ability [32]. From this population, we invited a number of individuals with PCS (as cases) and of symptom-free individuals after recovery (as controls) to undergo a comprehensive outpatient medical examination and clinical evaluation, including standardised and validated questionnaires, neurocognitive and cardiopulmonary testing and laboratory investigations. Based on estimates from experience in our and other PCS care centres and the literature available at the time of planning the study [33,34], we hypothesised that roughly half of the cases following the invitation would be persistent cases at the time of medical examination and expected that our clinical evaluation of patients with persistent PCS would result in an appreciable proportion of subjects with measurable organ dysfunction and pathology and show significant differences in at least one of the medical tests compared to the control group of individuals with continued recovery. We were also interested in markers and risk factors for more severe disease and its possible underlying pathophysiology.

## Materials and methods

### Study design and selection of participants

This study was a prospective, multi-centre, observational, nested case-control study. Participants with (cases) and without PCS (controls) were recruited from the _Epi_demiology of _Lo_ng _C_ovid (EPILOC) phase 1 non-interventional, population-based questionnaire study that included subjects aged 18–65 years who had tested positive for SARS-CoV-2 by PCR between October 1st, 2020 and April 1st, 2021, and whose infection had been notified (compulsory according to the German Infection Protection Act) to the responsible local public health authority (in four administratively and geographically defined regions in the Federal State of Baden-Württemberg in southwestern Germany). We estimated that most participants were infected with the wild type of SARS-CoV-2, that less than 15% of the cohort with B.1.1.7 (alpha) and less than 1% with B.1.351 (beta) [32].

The PCS case definition used was "general health or working capacity recovered to a level no more than 80% (compared to pre-COVID-19), and any new symptom (a list of 30 symptoms was provided, three additional symptoms could be added) of moderate to strong degree regarding impairment in daily life and not already present before the acute infection (excluding vomiting, nausea, stomach ache, diarrhoea, chills, fever)". Study participants who had recovered to 100% (of general health and work ability perceived in the time before acute infection) and reported no new symptoms of grade moderate-to-strong qualified as controls. Using these definitions, EPILOC phase 1 had categorised 28.5% of the 11,710 evaluable respondents as suffering from PCS (cases), whereas 38% of the respondents were considered as (PCS-free) recovered controls [32].

From these two groups, we invited participants into the phase 2 nested case-control study. Individuals qualifying as neither PCS case nor control were not invited. A total of 982 patients with PCS and 576 frequency-matched symptom-free recovered (control) participants (matched by sex- and age-group with a target sampling ratio of 1,000 cases to 600 controls) followed the invitation and underwent a comprehensive clinical evaluation at one of the four study sites (Fig 1). The unequal sampling ratio was based on the assumption that a significant number of phase 1 patients with PCS might have had recovered until presentation in phase 2, while we expected

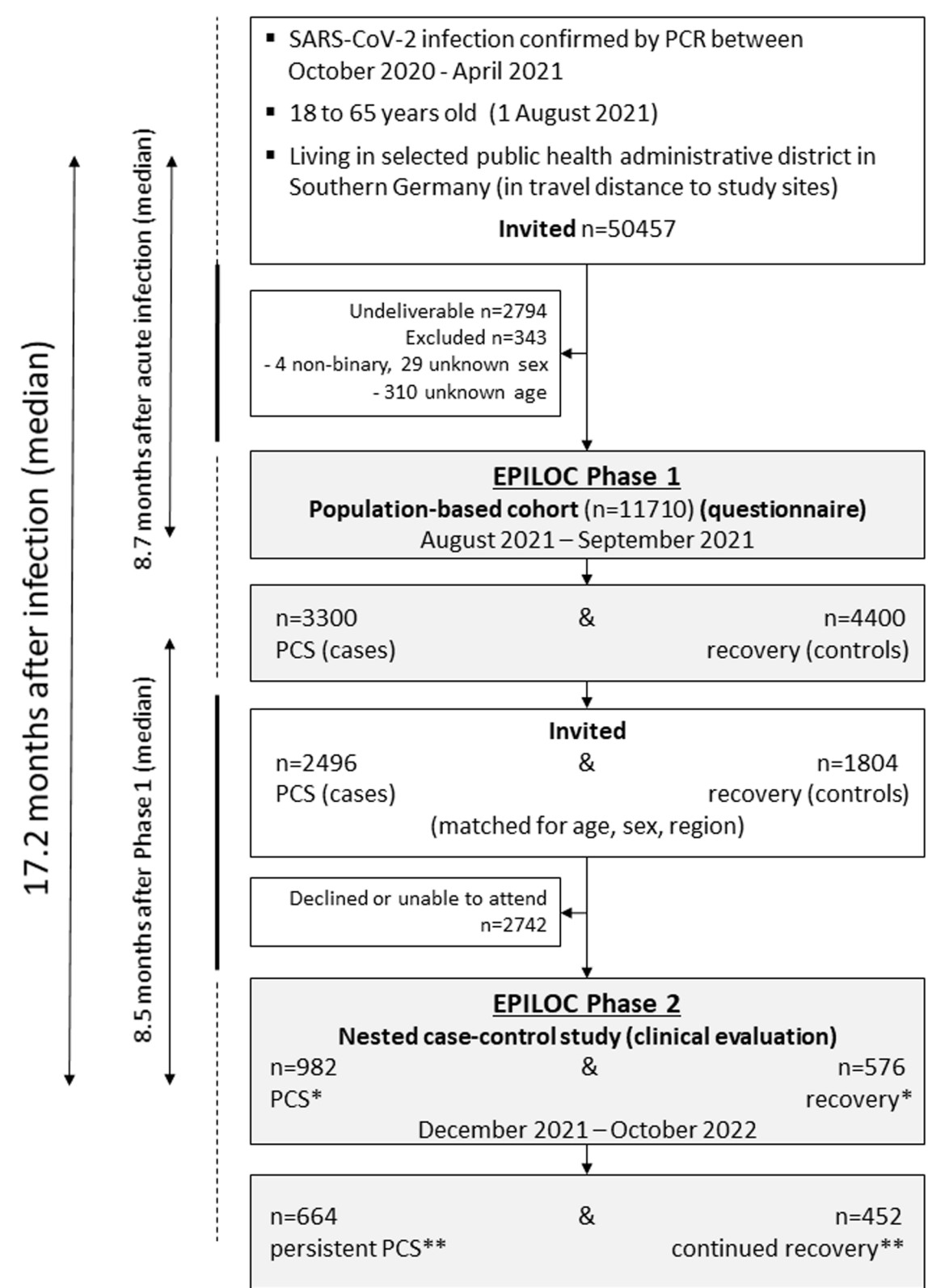

**Fig 1. Flow-chart covering EPILOC study phases 1 and 2.** NB *before (at phase 1) and **after clinical examination (in phase 2). The time from PCR-confirmed acute SARS-CoV-2 infection to phase 1 was 8.7 months (median), the time from phase 1 participation until clinical examination in phase 2 was 8.5 months (median), and the median time between acute infection and phase 2 was 17.2 months, ranging from 9.2 to 24.4 months.

that only a small number of symptom-free recovered participants might have developed new symptoms compatible with PCS at the time of the clinical evaluation in phase 2.

All study procedures and analyses were pre-planned and listed in a study handbook (only available in German language) with the exception of un-planned interim analyses of the tests for spike antigen in serum and for SARS-CoV-2 RNA in faecal samples; after negative results in both tests with samples from >100 patients with persistent PCS, the investigation of further samples (as initially planned) was discontinued (see below).

This study is reported as per the Strengthening the Reporting of Observational Studies in Epidemiology (STROBE) guideline (S1 STROBE Checklist).

## Data sources and measurements

Besides the information collected during the phase 1 study (see [32]), we again used data from a number of standardised questionnaires that included sociodemographic characteristics, lifestyle factors, SARS-CoV-2 vaccines received, medical history and current symptoms. The symptom questionnaire (see S1 Appendix) contained the same items as in phase 1 and asked for medical treatment of current symptoms, for the grade to which each symptom impaired daily life and activities ("how much do you feel impaired by this at the moment?") using a 4-point Likert scale (none, light, moderate or strong) and for the degree of general health and working capacity regained (compared with the time before the index infection). Based on this information, we defined participants either as having persistent (or improved) PCS or as individuals with continued recovery (or as recovered individual with worsening), using the same definition as in phase 1.

We evaluated individual symptoms, but also symptom clusters composed of highly inter-related individual symptoms as defined earlier after analysis of the phase 1 study results [32]. Details of the approach to define symptom clusters have previously been described [32].

**Clinical assessments.** Apart from taking the medical history, the study physician completed a modified-Medical Research Council Dyspnoea Scale (mMRC), asked for post-exertional malaise (PEM) and its duration [35], and clarified questions and responses to the questionnaires. The participants underwent a complete physical examination, including measurements of height, weight, heart rate (HR) at rest and blood pressure.

The maximal grip strength was recorded after three measurements of both hands with a digital hand dynamometer. Whole body composition was measured using a multi-frequency bioelectrical impedance analysis device and expressed as % body fat. Methodological details are included in S2 Appendix.

**Validated questionnaires.** Study participants were asked to fill validated questionnaires on sleep quality (Pittsburgh Sleep Quality Index [PSQI], Insomnia Severity Index [ISI], Epworth Sleepiness Scale [ESS]), fatigue (Chalder Fatigue Scale [CFQ-11]), health-related quality of life (Short Form-12 Health Survey [SF-12], assessing both physical and mental components), symptoms of depression (Patient Health Questionnaire 9 [PHQ-9]), anxiety (Generalised Anxiety Disorder 7 [GAD-7]), perceived stress (10-item Perceived Stress Scale [PSS-10]), subjective cognition ("Fragebogen zur geistigen Leistungsfähigkeit" [FLei]), and dysautonomia symptoms (Composite Autonomic Symptom Score 31 [COMPASS-31]). More details and references are given in S2 Appendix.

**Neurocognitive tests.** All participants were asked to undergo neuropsychological tests administered by trained clinical staff. The test battery included the Montreal Cognitive Assessment (MoCA), the Trail making test part B (TMT-B), and the Symbol Digit Modalities Test (SDMT) (S2 Appendix).

**Cardiopulmonary function tests.** We recorded resting 12-lead electrocardiograms (ECGs) and pulse oximeter measurements of peripheral oxygen saturation ($SpO_2$). Resting

echocardiograms were performed according to current guidelines, with determination of the left ventricular volume and ejection fraction (LV-EF), the ratio between early mitral inflow and mitral annular early diastolic velocities (LV-E/e'), the ratio of maximal early to late diastolic transmitral flow velocity (LV-E/A), and grading of diastolic dysfunction (for details see S3 Appendix).

Participants underwent cardiopulmonary exercise testing (CPET) using a ramp protocol on the cycle ergometer. Before CPET, spirometry was conducted to assess lung function with recording of the forced expiratory volume in one second (FEV1), and the forced vital capacity (FVC). During CPET, blood pressure, $SpO_2$ and ECG with HR were monitored. We evaluated the following CPET parameters: HR, oxygen uptake ($VO_{2max}$), breathing reserve (BR), respiratory exchange ratio (RER) and the slope of minute ventilation to carbon dioxide production ($VE/VCO_2$ slope). More details are included in S3 Appendix.

**Laboratory investigations.** Routine laboratory investigations included a rapid chromatographic immunoassay (for SARS-CoV-2 antigen in nasopharyngeal samples), blood cell counts, coagulation, clinical chemistry, levels of C-reactive protein (CRP), thyroid-stimulating hormone (TSH), glycated haemoglobin (HbA1c), N-terminal pro-brain natriuretic peptide (pro-BNP), classical pathway complement haemolytic activity (CH50) (determined for participants at two centres), immunoglobulin (Ig) G and IgM antibodies against cytomegalovirus (CMV), antibodies against SARS-CoV-2 nucleocapsid (N) protein and the S1 receptor binding domain of the viral spike glycoprotein, and others (see S4 Appendix for analytes and methods).

Cortisol, adrenocorticotropic hormone (ACTH) and dehydroepiandrosterone sulfate (DHEA-S) levels in frozen morning blood samples were measured centrally using standard methods (see S4 Appendix for details). Additional laboratory investigations in our central virology laboratory included the measurement of antibodies to Epstein–Barr virus (EBV) antigens, of spike antigen in serum (in a subgroup of individuals with persistent PCS and continued recovery), and SARS-CoV-2 RNA by reverse transcription polymerase chain reaction (RT-PCR) in faecal samples (see S4 Appendix for detailed methodologies).

## Statistical methods

Participant characteristics were analysed descriptively. Predictors of case-control status change from phase 1 to phase 2 were evaluated using logistic regression. Regression models were run separately for phase 1 cases and controls, and mutually adjusted odds ratios were calculated for improvement in cases (no longer fulfilling the case definition) and worsening in recovered individuals (no longer fulfilling the control definition).

Results of standardised questionnaires, neurocognitive tests, laboratory measurements, electrocardiographic, echocardiographic and spiroergometric parameters were presented as least square means. Due to a high correlation between PSQI, ISI and ESS, we present only the results for the PSQI instrument (see S2 Appendix).

We used analysis of covariance with adjustment for sex-age class combinations and university entrance qualification. Additional adjustments were made as indicated. Geometric instead of natural means are reported where appropriate. The area under the curve (AUC) for discrimination of persistent cases versus stable controls (excluding improved cases and worsened controls), based on logistic regression, is also reported. We did not use imputation, but missing observations were excluded in the specific analyses. Statistical procedures were performed with the SAS statistical software package (release 9.4 SAS Institute) or R version 4.3.2.

## Ethical approval

The study was registered with "Deutsches Register Klinischer Studien" (DRKS 00027362). All participants provided written informed consent. Ethical approval was obtained from the

respective ethical review boards of the study centres in Freiburg (21/1484_1), Heidelberg (S-846/2021), Tübingen (845/2021BO2), and Ulm (337/21).

## Results

### Baseline characteristics of the study participants

The study included 982 participants who were phase 1 PCS patients (cases) and 576 age- and sex-matched recovered subjects (phase 1 controls). As shown in S1 Table, the sex and age distributions were (as expected by design) similar in cases and controls. Most (65.8%) participants were female, and the mean age was 48 years. The mean time between phases 1 and 2 was 9.1 months for patients with PCS (range 3.0–14.2 months) and 8.4 months for recovered individuals (range 2.9–14.0 months). A similar proportion of patients with PCS versus recovered individuals experienced a secondary SARS-CoV-2 infection (23%) and almost all had been vaccinated against SARS-CoV-2 once or more times before phase 2 (S1 Table).

Differences between patients with PCS and recovered participants already known from the analysis of phase 1 data included the proportion of obese participants, smokers, pre-existing diseases, medical care (outpatient or inpatient versus none) for the earlier index acute SARS-CoV-2 infection (each higher among patients with PCS) and educational level (less frequent university entrance qualification among patients with PCS). Healthcare utilisation in the last 6 months prior to phase 2 examination (in particular regarding specialist physician consultation) and attending a rehabilitation programme were also much more frequent among patients with PCS. S1 Fig describes the probability of participation in the two groups of cases and controls by selected baseline characteristics.

### Risk for PCS persistence

Roughly, two-thirds (67.6%) of the 982 participants with PCS in phase 1 were considered having persistent PCS (according to our working definition) after the phase 2 clinical assessment. Most of the remaining participants with PCS phase 1 (30.1%) had improved until phase 2, but only very few (2.2%) were classified as completely clinically recovered (Fig 2). Conversely, the majority (78.5%) of symptom-free participants from phase 1 who participated in phase 2 were classified as having continued recovery, but almost one-fifth (18.9%) reported new symptoms (without fulfilling the PCS case definition), and 2.6% were classified as (new-onset) PCS cases (Fig 2). S2 Fig displays changes in the prevalence of the five main symptom clusters among the patients with PCS between phases 1 and 2. In the overall population, the net prevalence of all symptom clusters, except anxiety, depression or sleep disorder decreased, most prominent for smell and taste disorders (S2 Fig).

As summarised in Fig 2 (and detailed in S2 Table), factors associated with improvement (either to intermediate or control status) of PCS in an adjusted analysis were educational status (university entrance qualification), full-time employment (at phase 1), no medical care/treatment of the acute index infection (as a proxy for milder acute infection) and no (need for) specialist consultation within the last 6 months or participation in a post-COVID-19 rehabilitation program (the latter two probably a result of reverse causation). For recovered individuals, the odds of worsening until phase 2 were higher with lower educational status and after a secondary SARS-CoV-2 infection since phase 1. SARS-CoV-2 vaccination had no measurable association with improvement or worsening. Also, age, sex or the time between phases 1 and 2 was not statistically significantly associated with case-control status changes (S2 Table).

### Clinical evaluation of persistent PCS cases

In comparison of the characteristics of the four groups (persistent PCS, PCS improved, continued recovery, recovery with worsening) (Table 1), we found differences in educational

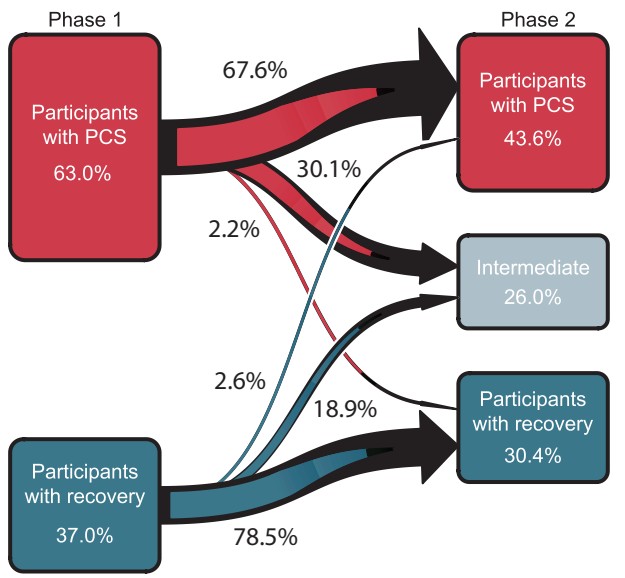

**Fig 2. Change in case-control status of study participants (***N* = 1,558**) between initial questionnaire survey (phase 1) and clinical examination (phase 2).** The time from phase 1 participation until clinical examination in phase 2 was 8.5 months (median). Factors associated with improvement of patients with PCS in phase 1 and with worsening among recovered participants in phase 1 were assessed for significance after calculation of ORs with mutual adjustment for the following variables: sex, age, university entrance qualification, marital status, medical treatment of acute infection, obesity (BMI ≥ 30 kg/m²), full-time employment (phase 1), time between phases 1 and 2 (per month), secondary SARS-CoV-2 infection since phase 1, two or more vaccine doses, (any) specialist consultation in the last 6 months, participation in a post-COVID-rehabilitation program (see S2 Table).

status, smoking, BMI (as well as obesity prevalence and body fat), medical care/treatment of the acute SARS-CoV-2 index infection and prevalence of comorbidities. The proportion of participants with obesity was highest in persistent PCS (30.2% compared with 12.4% in stable controls), and many more participants with continued recovery than with persistent PCS have had no medical care for their acute index infection, had obtained university entrance qualification and were never smokers (Table 1). We found a much higher current use of medication in patients with persistent PCS versus participants with continued recovery across all anatomical-therapeutic-chemical groups (S3 Table).

**Predominant symptoms, symptom clusters and symptom severity.** An analysis of the frequency of all reported symptoms with all degrees of impairment among participants with persistent PCS (S3 Fig) showed the predominance of individual complaints and symptoms that we summarise in the symptom clusters "fatigue", "neurocognitive disturbance", "chest symptoms", "smell or taste disorder" and "anxiety/depression/sleep disorder". As shown in S3 Fig, there were some differences in individual symptom prevalence and severity between female and male participants (with females being more affected—similar to findings in phase 1), and several individual symptoms were scored comparatively low regarding their grade of daily life impairment (e.g., dizziness, paraesthesia, confusion and chest pain). Abdominal symptoms, fever and chills, and skin problems were rare, similar to what we found in phase 1.

We next displayed the distribution of (case-defining, i.e., moderate-or-severe) predominant symptoms and symptom clusters among patients with persistent PCS versus the other subgroups, together with the scoring results from corresponding validated questionnaires either as proportions at relevant cut-offs (Table 2) or as adjusted average ratings (Fig 2). As shown in Table 2, fatigue, neurocognitive disturbance and chest symptoms were among the

**Table 1. Characteristics of the phase 2 study participants by case-control status.**

| | Persistent PCS | | PCS with improvement | | Recovery with worsening | | Continued recovery | |
|---|---|---|---|---|---|---|---|---|
| | N | Mean or frequency | N | Mean or frequency | N | Mean or frequency | N | Mean or frequency |
| Male, n (%) | 664 | 227 (34.2) | 318 | 122 (38.4) | 124 | 44 (35.5) | 452 | 153 (33.9) |
| Female, n (%) | | 437 (65.8) | | 196 (61.6) | | 80 (64.5) | | 299 (66.2) |
| Age at phase 1 (years), mean (SD) | 664 | 48.9 (12.1) | 318 | 46.3 (12.5) | 124 | 48.4 (11.9) | 452 | 48.5 (12.4) |
| Age class at phase 1 (years), n (%) | | | | | | | | |
| 18–29 | | 74 (11.1) | | 49 (15.4) | | 14 (11.3) | | 55 (12.2) |
| 30–39 | | 76 (11.5) | | 50 (15.7) | | 14 (11.3) | | 55 (12.2) |
| 40–49 | | 128 (19.3) | | 60 (18.9) | | 27 (21.8) | | 91 (20.1) |
| 50–59 | | 267 (40.2) | | 116 (36.5) | | 49 (39.5) | | 159 (35.2) |
| 60+ | | 119 (17.9) | | 43 (13.5) | | 20 (16.1) | | 92 (20.4) |
| University entrance qualification, n (%) | 664 | 257 (38.7) | 318 | 163 (51.3) | 124 | 60 (48.4) | 452 | 278 (61.5) |
| Full-time employment at phase 1, n (%) | 663 | 306 (46.2) | 318 | 194 (61.0) | 124 | 66 (53.2) | 451 | 223 (49.5) |
| Smoking status, n (%) | 662 | | 317 | | 124 | | 452 | |
| Current | | 52 (7.9) | | 20 (6.3) | | 10 (8.1) | | 17 (3.8) |
| Former | | 205 (31.0) | | 78 (24.6) | | 36 (29.0) | | 93 (20.6) |
| Never | | 405 (61.2) | | 219 (69.1) | | 78 (62.9) | | 342 (75.7) |
| BMI at phase 2 (kg/m²), mean (SD) | 662 | 28.0 (6.1) | 318 | 26.6 (5.5) | 124 | 26.1 (4.5) | 452 | 25.0 (4.5) |
| Obese (≥30 kg/m²), n (%) | | 200 (30.2) | | 64 (20.1) | | 25 (20.2) | | 56 (12.4) |
| Body fat (per cent), mean (SD) | 659 | 32.2 (10.6) | 316 | 29.3 (9.4) | 123 | 28.5 (9.0) | 452 | 27.4 (8.9) |
| >25% in men, >35% in women, n (%) | | 344 (52.2) | | 122 (38.6) | | 45 (36.6) | | 126 (27.9) |
| Treatment of acute SARS-CoV-2 infection, n (%) | | | | | | | | |
| No medical care | 655 | 341 (52.1) | 313 | 200 (63.9) | 123 | 108 (87.8) | 450 | 408 (90.7) |
| Outpatient care | | 258 (39.4) | | 92 (29.4) | | 12 (9.8) | | 37 (8.2) |
| Inpatient care (without ICU) | | 45 (6.9) | | 17 (5.4) | | 3 (2.4) | | 3 (0.7) |
| Intensive care | | 11 (1.7) | | 4 (1.3) | | 0 (0.0) | | 2 (0.4) |
| Comorbidities, n (%) | 664 | | 318 | | 124 | | 452 | |
| Cardiovascular disease | | 29 (4.4) | | 2 (0.6) | | 1 (0.8) | | 3 (0.7) |
| Chronic pulmonary disease | | 62 (9.3) | | 34 (10.7) | | 6 (4.8) | | 23 (5.1) |
| Diabetes mellitus | | 33 (5.0) | | 8 (2.5) | | 2 (1.6) | | 5 (1.1) |
| Cancer | | 13 (2.0) | | 3 (0.9) | | 1 (0.8) | | 4 (0.9) |

Note: BMI, body mass index; SD, standard deviation.

predominant symptom clusters in persistent PCS. We observed a large overlap of these three clusters, with a substantial proportion of patients with persistent PCS (26.8%) reporting moderate or severe symptoms in all three main symptom clusters (S4 Fig). The second largest overlap was the combination of fatigue and neurocognitive disturbance (prevalence, 20.1%). One or more of these three main symptom clusters affected the vast majority (90.4%) of participants with persistent PCS.

The frequency estimates for a given symptom or symptom cluster varied with more detailed questioning or rating, allowing a more valid estimation of severity. Fatigue as the most prevalent self-reported symptom cluster (based on reporting chronic fatigue or rapid physical exhaustion of moderate or strong grade in the symptom questionnaire), for example, had a prevalence among patients with persistent PCS of 67.6%, while the prevalence assessed with the CFQ-11 scale at a bimodal score >3 (earlier defined as a "fatigue case") or at a total score >19 was 92.1% and 69.8%, respectively. The prevalence of extreme fatigue (CFQ-11 total score >29) was relatively low among patients with persistent PCS (9.2%) (Table 2).

**Table 2. Prevalence of major symptom new clusters/symptoms and associated severity ratings according to validated questionnaires by case-control status at clinical examination in phase 2.**

| | Persistent PCS | | PCS with improvement | | Recovery with worsening | | Continued recovery | |
|---|---|---|---|---|---|---|---|---|
| | *N* | **Frequency** | *N* | Frequency | *N* | Frequency | *N* | **Frequency** |
| Fatigue/exhaustion/exertion intolerance, *n* (%) | | | | | | | | |
| Chronic fatigue and/or rapid physical exhaustion as moderate/severe symptom cluster | 661 | 449 (67.9) | 318 | 51 (16.0) | 124 | 15 (12.1) | 452 | 0 (0.0) |
| CFQ-11 bimodal score >3 | 649 | 598 (92.1) | 311 | 200 (64.3) | 122 | 44 (36.1) | 441 | 35 (7.9) |
| CFQ-11 total score >19 | | 453 (69.8) | | 76 (24.4) | | 15 (12.3) | | 6 (1.4) |
| CFQ-11 total score >29 | | 60 (9.2) | | 3 (1.0) | | 0 (0.0) | | 0 (0.0) |
| Fatigue with PEM lasting >14 h | 612 | 218 (35.6) | 300 | 15 (5.0) | 122 | 3 (2.5) | 450 | 0 (0.0) |
| ME/CFS-like (according to Canadian consensus criteria) | 649 | 75 (11.6) | 317 | 3 (1.0) | 124 | 2 (1.6) | 452 | 0 (0.0) |
| Neurocognitive disturbance, *n* (%) | | | | | | | | |
| Concentration difficulties as moderate/severe symptom | 663 | 416 (62.8) | 317 | 44 (13.9) | 124 | 15 (12.1) | 451 | 3 (0.7) |
| Memory difficulties as moderate/severe symptom | 664 | 360 (54.2) | 317 | 40 (12.6) | 124 | 11 (8.9) | 451 | 1 (0.2) |
| FLei memory subscore >19 | 662 | 360 (54.4) | 317 | 73 (23.0) | 122 | 11 (9.0) | 451 | 16 (3.6) |
| FLei attention subscore >19 | 643 | 281 (43.7) | 310 | 46 (14.8) | 123 | 9 (7.3) | 448 | 7 (1.6) |
| FLei total score >45 | 629 | 396 (63.0) | 309 | 80 (25.9) | 121 | 20 (16.5) | 445 | 18 (4.0) |
| Chest symptoms, *n* (%) | | | | | | | | |
| Chest pain, shortness of breath and/or wheezing as moderate/severe symptom cluster | 664 | 315 (47.4) | 318 | 42 (13.2) | 124 | 15 (12.1) | 452 | 0 (0.0) |
| Dyspnoea mMRC grade 1 | 656 | 274 (41.8) | 317 | 72 (22.7) | 124 | 18 (14.5) | 452 | 10 (2.2) |
| Dyspnoea mMRC grade 2 | | 48 (7.3) | | 5 (1.6) | | 2 (1.6) | | 0 (0.0) |
| Dyspnoea mMRC grade 3–4 | | 21 (3.2) | | 0 (0.0) | | 0 (0.0) | | 0 (0.0) |
| Anxiety/depression/sleep disorder, *n* (%) | | | | | | | | |
| Anxiety as moderate/severe symptom | 663 | 121 (18.3) | 318 | 18 (5.7) | 124 | 3 (2.4) | 452 | 0 (0.0) |
| GAD-7 score >9 | 658 | 244 (37.1) | 316 | 40 (12.7) | 123 | 8 (6.5) | 447 | 11 (2.5) |
| Depression as moderate/severe symptom | 664 | 176 (26.5) | 318 | 19 (6.0) | 124 | 10 (8.1) | 451 | 3 (0.7) |
| PHQ-9 score >14 | 646 | 148 (22.9) | 308 | 17 (5.5) | 122 | 6 (4.9) | 446 | 2 (0.5) |
| Sleep disorder as moderate/severe symptom | 664 | 327 (49.3) | 318 | 57 (17.9) | 123 | 33 (26.8) | 452 | 12 (2.7) |
| PSQI score >10 | 625 | 224 (35.8) | 307 | 40 (13.0) | 120 | 8 (6.7) | 439 | 8 (1.8) |
| ISI score >14 | 644 | 296 (46.0) | 313 | 55 (17.6) | 122 | 17 (14.0) | 443 | 11 (2.5) |
| ESS score >10 | 636 | 259 (40.7) | 310 | 76 (24.5) | 119 | 23 (19.3) | 443 | 31 (7.0) |

Note: CFQ-11 total score >19 or bimodal score >3: fatigue, CFQ-11 total score >29: extreme fatigue. FLei total score >45: subjectively impaired mental performance, FLei memory subscore >19: subjectively impaired memory, FLei attention subscore >19: subjectively impaired attention. mMRC grade 1: dyspnoea when hurrying or walking up a slight hill, mMRC grade 2: walks slower than people of the same age because of dyspnoea or has to stop for breath when walking at own pace, mMRC grade 3–4: stops for breath after walking 100 m or after a few minutes, or too dyspneic to leave house or breathless when dressing. GAD-7 score >9: moderate-to-severe anxiety. PHQ-9 score >14: moderate-to-severe depression. PSQI score >10: poor sleep quality. ISI score >14: insomnia; ESS score >10: excessive daytime sleepiness. For abbreviations and methods see text and S2 Appendix).

We also assessed the prevalence of fatigue with PEM lasting >14 h (35.6%) and of symptoms compatible with a myalgic encephalomyelitis (or encephalopathy)/chronic fatigue syndrome (ME/CFS)-like condition (11.6%). Interestingly, the frequency of individual symptoms (of any degree) among patients with PEM (lasting >14 hours) differed from those who had no PEM. Patients with persistent PCS and PEM had more symptoms than patients with persistent PCS without PEM. In particular, pain syndromes (chest pain, myalgia, joint pain, melalgia and headache), confusion and dizziness were more often reported by case patients with PEM (apart from fatigue and exhaustion) (S3 Fig). PEM was highly prevalent (>50%) among patients with persistent PCS who reported symptoms from all three dominant clusters (fatigue, neurocognitive disturbances and chest symptoms) (S5 Fig).

Neurocognitive impairment remained the second most frequent symptom cluster (per symptom questionnaire) after fatigue in patients with persistent PCS, which correlated well with the FLei questionnaire results (Table 2). Dyspnoea was most often non-severe when assessed with mMRC grading (Table 2). The prevalence of mMRC grade 1 dyspnoea among patients with persistent PCS was 41.8%, and dyspnoea of grade 2 or more was seen in 10.5%. Symptoms of anxiety, depression and sleep disorders (that had earlier been classified as a single cluster of highly interrelated symptoms) were also much more prevalent among patients with persistent PCS than among participants with continued recovery. The average scores of CFQ-11, FLei, GAD-7, PHQ-9 and PSQI differed substantially and consistently between the subgroups, and all these instruments discriminated participants with versus those without persistent PCS very well, with the CFQ-11 having the highest AUC (>0.90) (Fig 3).

**Symptoms of dysautonomia.** As shown in Fig 3, the average COMPASS-31 score among patients with persistent PCS was 13 compared with <2 among individuals with continued recovery, and the proportion of patients with persistent PCS with a score >19 (suggesting moderate or severe dysautonomia) was 40.7%. Almost half of the individuals with persistent PCS (49.7% compared with 7.5% of individuals with continued recovery) indicated that they experienced weakness, dizziness, light-headedness or difficulty thinking after standing up from sitting or lying down, suggesting orthostatic problems.

**Perceived stress and health-related quality of life.** As a measure of stress and health-related quality of life, we used the PSS-10 instrument (scoring from 0 to 40) and the commonly used SF-12 questionnaire with its physical and mental component summary scores, assessing general health and well-being, including the perceived impact of any illnesses or adverse condition on a broad range of functional domains. As shown in Fig 3, all three scores discriminated well between individuals with persistent PCS and those with continued recovery and had similarly high AUCs >0.8. The differences in the average scores between patients with persistent versus improved PCS and between individuals with continued recovery versus initial recovery with worsening showed a similar pattern as the other instruments. A direct comparison of the current SF-12 results for both components among patients with PCS with the results obtained earlier in the same individuals (at phase 1) indicated no improvement in health-related quality of life, with mean changes in the physical subscale of −0.51 (95% confidence interval [CI] [−1.13, 0.10]), and in the mental subscale of −0.92 (95% CI [−1.68, −0.16]), respectively.

**Neurocognitive testing.** The results of the three neurocognitive tests are depicted in Fig 3. In adjusted analysis, the mean MoCA score was significantly lower among patients with persistent PCS compared with the other groups, and the proportion of participants with a MoCA score below 26 (suggesting mild-to-moderate cognitive impairment) was 33.3% among patients with persistent PCS versus 18.9% among participants with continued recovery, respectively. Similar patterns were seen with the two other tests, SDMT (assessing impaired attention, concentration and speed of information processing) and TMT-B (to screen executive dysfunction). Although the mean differences between cases and controls were large, the discrimination in adjusted analysis between the two groups; however, was relatively poor for each test (AUCs 0.67 compared to 0.63 without neurocognitive testing). Further adjustment for CFQ-11 and PHQ-9 attenuated the association with MoCA to some degree, with differences for participants with persistent PCS versus participants with continued recovery losing statistical significance ($p = 0.0672$). However, the additional adjustment had little effect on the association with SDMT and TMT-B ($p = 0.0086$ and $0.0008$).

**Grip strength and cardiopulmonary function tests.** The mean maximal handgrip strength was 40.2 kg among patients with persistent PCS, significantly lower than among participants with continued recovery (42.5 kg) (Fig 4). As expected, grip strength was lower

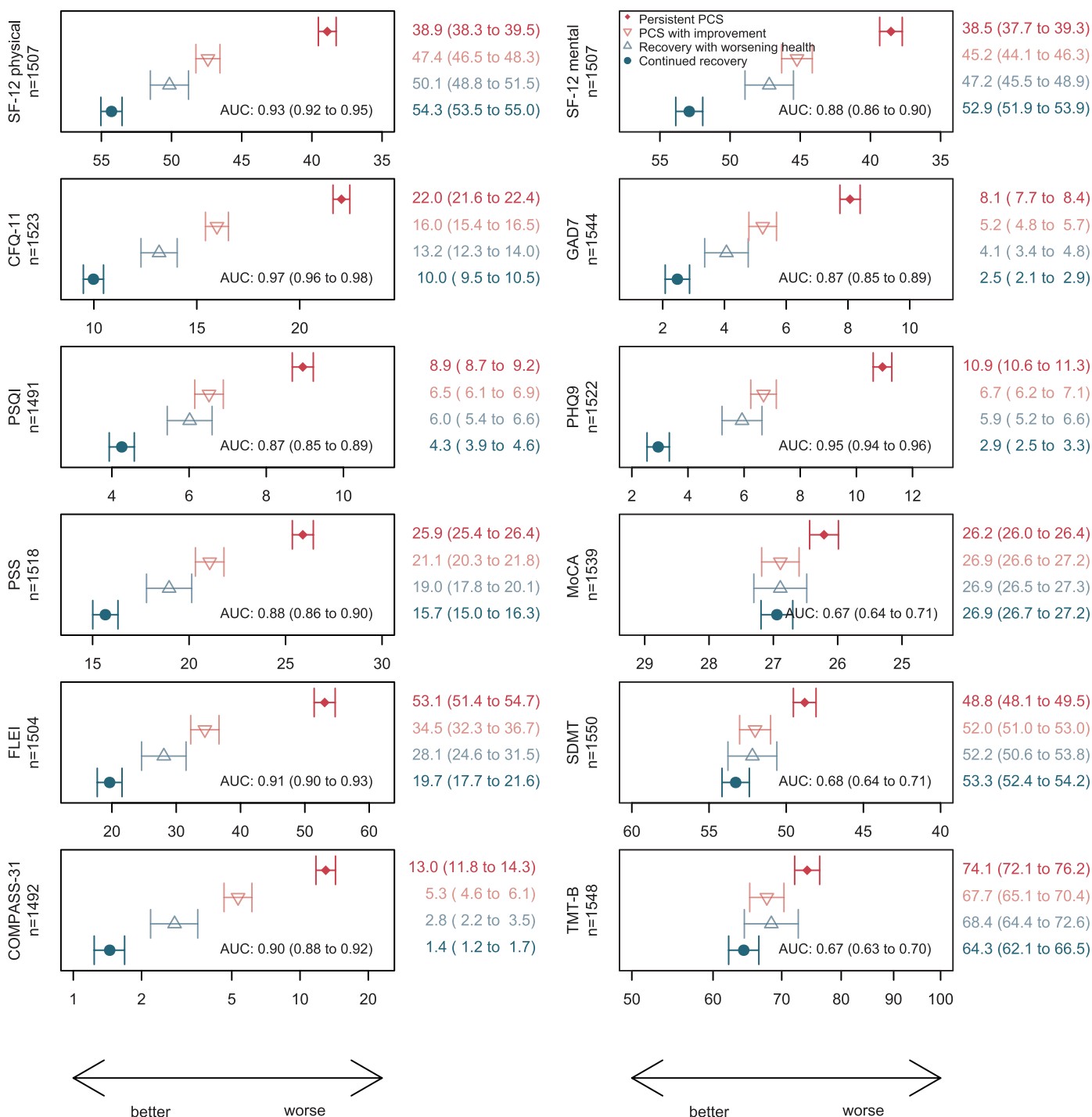

**Fig 3. Means (geometric mean for COMPASS-31 and TMT-B) of self-reported health outcomes and neurocognitive tests (with 95% CI) by case-control status at clinical examination in phase 2, adjusted for sex-age class combinations, study centre and university entrance qualification.** The reported area under the curve (AUC) for persistent PCS vs. continued recovery by the respective instrument also includes sex-age class combinations and university entrance qualification. The AUC for sex-age class combinations, study centre and university entrance qualification alone was 0.64. For comparability, the x-axis is scaled from mean −1 SD to mean +1 SD for all panels. Abbreviations: PSQI, Pittsburgh Sleep Quality Index; CFQ-11, Chalder Fatigue Scale; SF-12, Short Form-12 Health Survey; PHQ-9, Patient Health Questionnaire 9; GAD-7, Generalised Anxiety Disorder 7; PSS-10, Perceived Stress Scale; Flei, "Fragebogen zur geistigen Leistungsfähigkeit" (subjective mental performance questionnaire); COMPASS-31, Composite Autonomic Symptom Score 31; MoCA, Montreal cognitive assessment scale (points); SDMT, Symbol Digit Modalities Test (number of correct symbols); TMT-B, Trail making test part B (time in seconds).

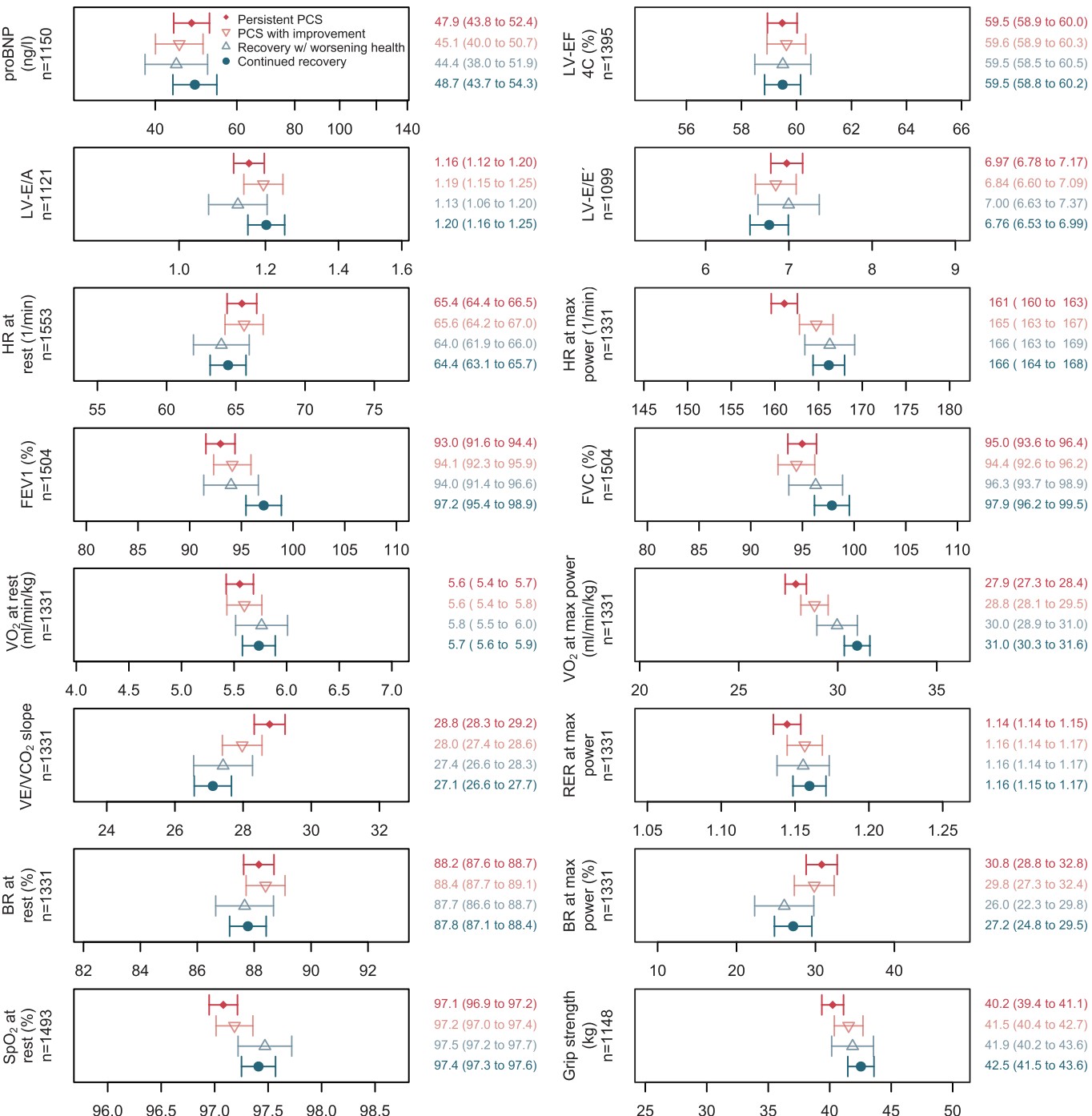

**Fig 4. Cardiopulmonary function indicators and grip strength (means with 95% CI) by case-control status at the clinical examination in phase 2, adjusted for sex-age class combinations, study centre, university entrance qualification, BMI, smoking status and use of beta blocking agents.** Cardiopulmonary exercise testing could be completed in 1,331 participants (87.2% of participants with continued recovery; 83.7% of patients with persistent PCS). For comparability, the *x*-axis is scaled from mean −1 SD to mean +1 SD for all panels. Abbreviations: pro-BNP, N-terminal brain natriuretic peptide; LV-EF, left ventricular volume and ejection fraction; LV-E/e′, ratio between early mitral inflow and mitral annular early diastolic velocities; LV-E/A, ratio of maximal early to late diastolic transmitral flow velocity; FEV1, forced expiratory volume in one second; FVC, forced vital capacity; SpO$_2$, peripheral oxygen saturation; HR, heart rate; VO$_{2max}$, oxygen uptake; BR, breathing reserve; RER, respiratory exchange ratio; VE/VCO$_2$ slope, slope of minute ventilation to carbon dioxide production.

in women than men (30.9 versus 50.7 kg) and associated inversely with body fat and BMI ($r = -0.06$, 95% CI [−0.12, −0.01] and $r = -0.07$, 95% CI [−0.12, −0.01]).

As depicted in Fig 4, left ventricular function (including LV-EF, LV-E/e′ and LV-E/A) and pro-BNP blood levels were not different between the groups. We observed a higher prevalence of diastolic dysfunction grades 1 and 2 among patients with persistent PCS compared with participants with continued recovery (30.9% versus 21.9%) (S4 Table). The difference; however, was not statistically significant after adjustment for sex-age class combinations, study centre, university entrance qualification, BMI and smoking status. Also, we did not observe differences between the subgroups in the mean values for resting HR and BR (Fig 4), respiratory rate (adjusted mean: 16.2–16.7 per minute for all subgroups, $p = 0.71$) and systolic and diastolic blood pressure (adjusted systolic mean: 128–131 mm Hg for all subgroups, $p = 0.88$; diastolic mean: 80–81 mm Hg for all subgroups, $p = 0.21$).

Differences were observed for FEV1 and FVC, $SpO_2$ at rest, and several CPET derived variables (Fig 4). Values for FEV1 ($p < 0.0001$), FVC ($p = 0.0011$) and $SpO_2$ ($p = 0.0001$) were lower among subjects with persistent PCS (versus participants with continued recovery), but the differences were small, and the proportion of persons with FEV1/FVC <0.70 was similar in both participants with or without PCS (10.3% versus 9.6%) (S4 Table).

In CPET, patients with persistent PCS achieved a lower maximal power with lower HR than the participants of the other subgroups, but RER values at the end of CPET were similar and well above 1.05, indicating exhaustion and attaining $VO_{2max}$. Also, the median values of the Borg CR10 scale were similar for persistent case patients and the control group (median = 8). The most relevant and significant CPET differences between patients with persistent PCS and participants with continued recovery were observed for $VE/VCO_2$ slope (higher values in persistent PCS) and $VO_{2max}$ (lower values in persistent PCS) (Fig 4). The proportion of patients with persistent PCS with $VO_{2max} < 85\%$ of target value (suggesting reduced exercise capacity possibly due to deconditioning or peripheral muscle limitations) was significantly greater than that of stable control subjects in adjusted analyses (35.3% versus 8.4%) (S4 Table). Similarly, the differences in the proportion of participants with $VO_{2max}$ below defined thresholds for males and females were substantial and highly significant between persistent case patients and participants with continued recovery S4 Table). Furthermore, the significant difference in the mean $VE/VCO_2$ slope (28.8 versus 27.1) (Fig 4) between patients with persistent PCS and participants with continued recovery corresponded to higher proportions of the participants with persistent PCS (versus the participants with continued recovery) with $VE/VCO_2$ slope values >30 (34.9% versus 18.5%) or >34 (13.5% versus 4.1%) (S4 Table).

We explored a possible overlap of objective signs of cognition deficits and reduced cardio-respiratory capacity within the persistent PCS case-patient population (S5 Table). The proportion of participants with MoCA ≤25 and SDMT <36 increased with increasing $VE/VCO_2$ slope, and there were more participants with SDMT <36 among patients with persistent PCS and poor $VO_{2max}$ (<85% predicted), but there were no such results for TMT-B, and patients with persistent PCS with differences in the Tiffenau test did not differ in their cognitive test performances (S5 Table).

**Laboratory investigations.** Besides pro-BNP (see above and Fig 4), we measured complete blood counts, several blood levels including CRP, lactate dehydrogenase, ferritin, liver and renal function and coagulation markers (D-dimer, von Willebrand factor antigen and activity), TSH, cortisol, ACTH, DHEA-S, HbA1c, 25-hydroxy-vitamin D3, CH50 and others. After adjustment for sex-age class combinations, study centre, university entrance qualification, BMI and smoking status, we found no significant differences between patients with persistent PCS and individuals with continued recovery in any of these laboratory investigations (S6 and S7 Figs and S6 Table and S4 Appendix). Notably, levels of CRP, HbA1c

and D-dimers were significantly higher in patients with persistent PCS than in the other subgroups ($p = 0.004$, $p = 0.001$, $p = 0.01$, respectively) before adjustment for BMI and smoking.

We did not observe significant differences in the prevalence of N SARS-CoV-2 antibodies ($p = 0.82$) or in the level of antibodies against SARS-CoV-2 S1 antigen (S8 Fig). Also, positivity rates for antibodies against CMV and several EBV antigens (viral capsid antigen [VCA], EBV nuclear antigen [EBNA] and early antigen D [EA-D]) did not differ significantly between groups (S7 Table). The proportion of study participants with EBV serology indicative of reactivation was 13% (194 of 1,468 seropositive participants). However, we detected no elevated risk for EBV reactivation among patients with persistent PCS or recovered individuals reporting new symptoms between phases 1 and 2 (S7 Table). We additionally looked at EA-D and EBNA IgG antibody levels in participants with evidence for EBV reactivation but did not observe differences between patients with persistent PCS with or without PEM and individuals with continued recovery (S9 Fig).

All study participants were negative for SARS-CoV-2 antigen in oropharyngeal swabs by a rapid antigen assay at presentation. Using an ultrasensitive antigen ECL assay, we could not detect SARS-CoV-2 spike antigen in plasma samples from a subgroup of 100 participants with persistent PCS and 100 persons in the control group. Also, RT-PCR for SARS-CoV-2 RNA was negative in all tested stool samples from a similar subgroup of 156 patients with persistent PCS and 103 participants with continued recovery (see also S4 Appendix), allowing to state with a certainty of 95% that the true PCR positivity prevalence in patients with persistent PCS 17 months after infection is less than or equal to 1.9%.

## Sensitivity analyses

The results of several sensitivity analyses (pre-existing illness/comorbidity, obesity, PEM, medical care of the index acute infection) are presented in the Supporting information figures in S1 Sensitivity Analyses. The general patterns persisted as described above, and the differences in the validated questionnaire scores, in neurocognitive as well as in cardiopulmonary tests that were significant in the full analysis set, remained significant. The odds of finding abnormal neurocognitive and cardiopulmonary test results were higher for female than for male participants with persistent PCS, but the differences were significant only for the TMT-B test (S10 Fig).

We also show that in the subpopulation of participants without pre-existing diseases and comorbidity, the changes between phases 1 and 2 in the prevalence of main symptom clusters were similar to those observed in the full analysis (S2 Fig). When participants with persistent PCS were stratified according to PEM (lasting >14 h), the burden of symptoms and complaints as reported and as assessed by validated questionnaires was much higher among patients with versus those without PEM symptoms, including sleep problems, depression and anxiety, perceived stress and subjective cognition impairment, fatigue and dysautonomia (S3 Fig and Fig G in S1 Sensitivity Analyses). The analysis of neurocognitive testing also showed PEM to be associated with substantially worse results (Fig G in S1 Sensitivity Analyses), particularly in the SDMT which assesses cognitive processing speed. However, participants with persistent PCS without PEM still had significantly worse results in all three tests than participants in the control group. Patients with persistent PCS and PEM also showed reduced handgrip strength, lower SpO$_2$, lower peak HR, higher values for VE/VCO$_2$ slope and reduced VO$_{2max}$ when compared with patients without PEM (Fig H in S1 Sensitivity Analyses), and the proportion with VO$_{2max}$ < 85% of target value was higher (41.0% versus 32.5% in persistent PCS with versus without PEM). Several other variables of cardiopulmonary function differed between the two subgroups (Fig H in S1 Sensitivity Analyses), although some showed only small clinically non-relevant differences (e.g., LV-E/A).

## Discussion

In this nested population-based case-control study, we found persistence of symptoms and impairments in two-thirds of patients with PCS after more than 1 year following acute SARS-CoV-2 infection. The comprehensive medical evaluation and comparison of individuals with persistent PCS with a control group of age- and sex-matched symptom-free convalesced persons demonstrated that many of the patients with persistent PCS had objective signs of cognitive deficits and reduced exercise capacity. Apart from observing large and discriminant differences in standardised measures of fatigue, neurocognitive disturbance, sleep quality, perceived stress, depression, anxiety, dysautonomia and quality of life, we detected significant differences between participants with persistent PCS and participants with continued recovery in MoCA, SDMT and TMT-B tests, in grip strength, $VO_{2max}$, $VE/VCO_2$ slope and a few other exercise capacity measures, and this finding was independent of age, sex, BMI and education (as probably the most significant potential confounding factors) and other variables. In contrast, laboratory tests (including inflammatory and coagulation markers) or resting echocardiographic results were not different after adjustment for covariates and were unable to discriminate cases from controls. These observations appear important since, unlike in many other studies, we included only adults in working age, and most study participants did not have medical treatment and were not hospitalised for their acute SARS-CoV-2 infection. Also, the initial population-based survey from which the participant population for the present study was retrieved had been performed 6–12 months following acute infection and thereby excluded persons with post-acute symptoms in the sense of delayed convalescence.

In the majority of participants who had developed PCS 6–12 months after COVID-19, symptoms and complaints persisted, and most of the 32% of the patients who reported an improvement at follow-up did not fully recover. In a recent Swiss study [36], the proportion of persons returning to a normal health status between 6 and 24 months after acute infection was roughly 25%, while the rate of improvement of symptoms associated with PCS was 37%. In another Swiss study [37], the proportion of patients with PCS and improvement between 7 and 15 months after acute infection was 48%. In both studies as well as in other work [38–40], there was a tendency of disease chronification beyond 6–12 months after acute infection, and our current findings support these observations. We saw some differential evolution of the predominant symptom clusters between phases 1 and 2. Fatigue, chest symptoms and smell/taste disorders showed a net decrease in prevalence over time. In contrast, the rate of improvement of the cognition and the depression/anxiety/sleep disorder clusters was similar to the rate of aggravation, resulting in only minor changes in the net prevalence. Others have also observed a tendency for more persistence of neurocognitive disturbances rather than other symptom clusters [16,41–47]. Stratified longitudinal analyses with objective measures are needed to better evaluate chronicity and prognosis of cognition deficits or other organic impairments, and such studies may benefit from advanced methods for defining different recovery clusters and multi-parameter modelling with validation across different cohorts [7,48–51].

Interestingly, risk factors for non-improvement of case status in the present study included lower educational status, and this was complemented by the finding of lower educational status as a risk factor for worsening health among initially recovered persons—besides secondary SARS-CoV infection. In the study reported by Hartung and colleagues [52], lower education was associated with cognitive non-recovery but not with persisting fatigue. In a large online survey [47], lower educational status was associated with worse symptom scores at all-time points post-infection. In our previous phase 1 study, lower educational status was already found to be associated with symptomatic disease at 6–12 months post-infection, and a similar association has been reported from two large US-cohorts [53]. We cannot exclude that

sampling bias accounts for these observations. Educational status, in general, is strongly associated with many underlying social, economic, lifestyle and behavioural factors. Which factors behind the educational status variable accounts most for the improvement/worsening effects is not known. Employment, obviously, was an independent factor for case-control status change between the two phases. The fact that we found cases without recent specialist consultation and without participation in rehabilitation between phases 1 and 2 to be more likely to improve, probably reflects a less severe acute and post-acute illness with a better prognosis (i.e., reverse causation).

An important finding was that post-acute vaccination against SARS-CoV-2 did not appear to be associated with PCS improvement. Several studies have shown a decreased PCS prevalence after vaccination, but it was often unclear whether one or more of the vaccine doses were in fact administered after illness onset [11]. Also, many studies were retrospective and did not adjust for confounders. In the study reported by Tran and colleagues [54], in which vaccine recipients with PCS were propensity score matched to non-vaccinated individuals with PCS and observed for 4 months, there were positive associations of (a first) vaccination with fewer symptoms, less severity and remission of PCS. In our study, the proportion of post-infection vaccine recipients was large. Almost all participants had already received their first vaccine before phase 1 (without measurable effects on symptom prevalence and severity), and many had received their second or booster doses between phases 1 and 2. As almost all had been vaccinated, it is difficult to ascertain a relationship between vaccination and recovery from PCS.

Symptom ratings and questionnaire data consistently showed that fatigue and cognitive disturbance were the most prevalent health problems (>60% for each cluster) among individuals with persistent PCS, a finding confirming the results of other studies with a similar follow-up time [43]. Of note were the large overlap between self-reported fatigue, cognition problems and chest symptoms and the strong correlation of various symptom ratings with health-related quality of life scores. Extreme fatigue and symptoms compatible with ME/CFS affected approximately one-tenth of the patients with persistent PCS, while PEM lasting >14 h was reported by 36% and was associated with worse scores in all questionnaires, but also in cognitive and cardiopulmonary exercise tests. This underscores the usefulness of including the history and duration of PEM when exploring patients with possible PCS [55,56]. Using the full set of DePaul questionnaire items, estimates for PEM might have been higher. In a Swiss cohort, PEM was observed in 48% of PCS patients, but in that study, fewer subjects (6%) fulfilled the criteria for ME/CFS [57]. A prevalence of 45% for PEM was observed in a Dutch cohort of PCS patients [58].

Cognitive disturbance was the second most frequent symptom cluster, with concentration problems being slightly more often reported than memory problems. A similar observation independent of the time after acute infection has also been made in a large online survey among subjects with complaints for at least 3 months after infection [47]. In a large claims data network analysis of neurologic and psychiatric sequelae, Taquet and colleagues [59] found that risks of cognitive deficits, dementia, psychotic disorders and epilepsy/seizures remained increased over a 2-year follow-up period after SARS-CoV-2 infection, which was unlike the risks of (other) common psychiatric disorders that rapidly returned to baseline. Other studies also reported persisting or increasing cognition or concentration problems with generally decreasing rates of other symptoms and physical health over time [16,41–47]. A memory questionnaire study found worse memory problems up to 3 years after acute infection (when compared to uninfected controls) [60], and a recent elegant study showed reaction time slowing with increasing time after acute SARS-CoV-2 infection [61]. Taken together, these findings and the results of the present study indicate that cognition problems might, in

fact, tend more to chronicity than other health problems of PCS patients. Reports of lower prevalence (22%–32%) of cognitive disturbances in meta-analyses may be due to differences in sample composition (more patients hospitalised during acute infection) and shorter follow-up times.

Sleep disorder, in particular insomnia, was another frequent complaint among cases. Pooled data of previous studies on >15,000 participants revealed a prevalence of 40%–50% for sleep disorder among individuals with PCS [62], which is comparable to our data. The importance of pre-pandemic healthy sleep to prevent PCS has been demonstrated by us and others [63,64]. It will be interesting to explore whether poor sleep quality remains a risk factor for continued non-recovery from PCS. Symptom reports and rating data on depressive and anxiety symptoms generally fit in the meta-analyses on neuropsychiatric manifestations in PCS [62,65].

We note that most of the routine clinical examination results and laboratory measurements did not discriminate between persistent cases and controls, including resting left ventricular systolic and diastolic function as well as the Tiffeneau test. These findings are essentially in line with the results of many other groups [48,66–70]. Small differences in values after crude analyses were no longer statistically significant after adjustment, in particular for BMI, smoking status and study site. D-dimer levels, for example, were slightly elevated among individuals with persistent PCS, but the differences were not significant in adjusted analyses, a result similar to those seen in earlier reports [67,71,72]. Because several studies suggested hypocortisolism as a possible explanation for PCS in at least some patients [20,73,74], we included blood levels of cortisol, ACTH and DHEA-S in our analysis. However, we could not find significant differences between persistent PCS and controls, suggesting a low likelihood of subacute or chronic adrenal insufficiency as a major contributing factor for PCS symptoms. Other recent studies also failed to identify differences in cortisol levels between PCS patients and several control groups [23,75,76]. Furthermore, we were not able to detect differences between persistent PCS and controls in complement turnover, a hypothesis recently raised in a number of studies [77,78]. We did; however, screen only for differences in CH50, but not for individual complement component blood levels.

Serological investigations indicated that the SARS-CoV-2 spike S1 antibody levels in our cohort were essentially driven by vaccination rather than being associated with PCS (as reported by Klein and colleagues),[20] and we did not find a significant association between elevated EA-D IgG antibodies (suggesting EBV reactivation) and PCS in an adjusted analysis. Previous data on this issue have been conflicting, with studies reporting [20,79,80] or failing to find [81,82] EBV reactivation markers associated with PCS. It has to be kept in mind that EA-D IgG antibody levels rise early after active viral replication and typically remain positive for only 3–6 months, while our samples were collected >12 months after acute SARS-CoV-2 infection which does not exclude a role of acute or early post-acute reactivation. However, we also did not observe increased levels of IgG antibodies against EBNA, which has been suggested as a longer-lasting surrogate for EBV reactivation and have previously been associated with neurocognitive disturbances in patients with PCS [83].

SARS-CoV-2 persistence has been proposed as another mechanism in non-recovery and PCS development. However, in our analysis, we did not observe antigen positivity in nasopharyngeal specimens, PCR positivity in stool samples, or viral antigen in plasma, which argues against persistent virus replication as a driver of PCS. The prevalence of viral persistence in non-invasive biospecimens from individuals with PCS as measured by a variety of methods has also been low in previous studies [82,84–86], with the exception of two small studies that showed spike antigenemia in >60% of patients with PCS some of whom were also PCR-positive in plasma samples [87,88], and a study reporting S1 protein persistence in monocyte

populations of patients with PCS up to 15 months post-infection [89]. A recent large study demonstrated that throat swab samples in a subgroup of patients with PCS and repeated PCR positivity in the early post-acute phase became negative beyond 3 months after acute infection [90]. Both spike and N protein were detected in plasma samples of 10 out of 100 patients with severe illness for at least 3 months (exact times not stated) after COVID-19 [82], but there was no apparent link between detectable antigen and symptoms. No viral RNA was detected in stool samples taken >300 days after acute infection, while prolonged shedding was associated with gastrointestinal symptoms but not PCS. In an exploratory study [91], four out of five subjects with a variety of symptoms had positive SARS-CoV-2 RNA detected in rectal biopsies obtained between days 158 and 676 after acute infection [92]. So far, very few patients with PCS and symptoms >12 months have been investigated for viral antigen/protein and/or RNA persistence [93], and an association between viral persistence and PCS remains an unproven hypothesis.

Neurocognitive testing showed significant group differences, indicating cognition deficits concerning attention and executive functioning, with problems in divided attention (TMT-B) and lower processing speed (SDMT) in patients with persistent PCS, and this finding appeared to be independent of pre-existing illnesses. One-third of the participants with persistent PCS (versus 18.9% among recovered participants) showed MoCA values < 26, which is slightly higher than observed in previous studies [61,94]. The mean value among participants with persistent PCS was 26.2 (25.8 in cases with PEM) compared with 26.9 among participants with continued recovery (and similar values in the other two groups). This small albeit significant difference may at least partly be related to the fact that the MoCA has limited specificity as a test originally designed to detect mild cognitive impairment among the elderly.

Impaired executive functioning and reduced processing speed, as observed here in persistent PCS is in agreement with a report of similar deficits observed in a large registry cohort [15] of COVID-19 patients followed up with multi-domain cognitive assessment, with pronounced cognitive slowing in 270 patients from two PCS cohorts [15,61], and with attention and executive function deficits in a comprehensive cognitive assessment of patients with PCS after mostly mild initial disease [95]. Although the cognitive findings described in the present study may be insufficient as a diagnostic aid to differentiate cases from controls because of the small-to-medium effect sizes, the data can help to better understand the nature of cognitive impairments in PCS. Controlling the group differences in cognitive test results for fatigue or depressive symptoms attenuated the association of the case status with the MoCA to some degree, but had little effect on the SDMT and TMT-B group differences, indicating that depressive mood and fatigue alone cannot explain the reduced performance in cognitive tests. This is in accordance with previous data [96]. Taken together, the information so far supports the concept of different pathomechanisms with regard to depression and cognitive disorders in PCS.

An impaired physical exercise capacity with reduced handgrip strength (or 6-min walk test) and reduced $VO_{2max}$ appear to be hallmark signs of PCS. Both measures were significantly different between patients with persistent PCS and participants with continued recovery in the present study. A reduced $VO_{2max}$ (<85% predicted) was observed in 35% of the persistent PCS patients, which is comparable to the prevalence found recently in other studies [97]. Similar to earlier observations [97–101], we also found a lower peak HR among patients with persistent PCS, while $RER_{max}$ and the rate of perceived exertion were similar. Taken together, these findings are compatible with deconditioning as a contributor to the impaired performance capacity [102], but muscular dysfunction/myopathy possibly due to mitochondrial lesions, may be an alternative explanation and additional mechanism. Ventilatory inefficiency is likely to be another contributing factor. Breathlessness as a moderate-to-severe symptom was reported by

almost 50% of patients with persistent PCS who also had significantly higher $VE/VCO_2$ slope values than stable control subjects. Other investigators have also found such differences in $VE/VCO_2$ slope between cases and controls [70,103,104]. The prevalence among PCS patients of a $VE/VCO_2$ slope >30 (increased) or >34 (abnormal) in our study was substantial (35% and 14%, respectively), greater than among recovered persons and similar to the proportions reported by Sørensen and colleagues. Even subtle differences in $VE/VCO_2$ slope may impact cardiorespiratory symptom severity after exercising [99,101]. Besides hyperventilation, erratic breathing with high variability in tidal volume and breathing frequency was described in quite a number of patients with PCS [105–109]. However, there is no universal gold standard for diagnosing dysfunctional breathing, and the present study did not include systematic screening for erratic breathing. Again, dysfunctional breathing would also be compatible with respiratory muscular dysfunction.

In accordance with previous data [98,103,110], the normal systolic function in the resting echocardiography in patients with persistent PCS described in the present study suggests that the reduced performance capacity is not caused by central cardiac limitation. Also, bronchial obstruction does not seem to be a cause for the hyperventilatory response to exercise since Tiffeneau tests were similar across all subgroups and BR was not exhausted. The (slightly) reduced FVC among cases (95.9% versus 99.1% for controls) is small but noteworthy. Longitudinal studies assessing FVC changes over time after SARS-CoV-2 infection produced conflicting results [70,111,112], while several cross-sectional studies have shown reduced lung volume associated with persistent symptoms [70,98,113,114]. In a study with patients hospitalised for acute infection [115], reduced FVC at 4 months correlated with increased findings in chest tomographs, reduced lung diffusion capacity, lower $SpO_2$, reduced exercise capacity, more fatigue and lower quality of life. The reason for the lower lung volume in our patients with PCS who had typically not been hospitalised may be respiratory muscle weakness [109,116,117], which remains to be further elucidated. There has been no clear evidence for an impairment of lung diffusion capacity among patients with initially mild acute infection [99,118]. Lung diffusion capacity was not measured in the present study. However, $SpO_2$ at cessation of exercise was not different between groups, making such a hypothesis in our study participants unlikely. Finally, we cannot exclude that the CPET results were affected by a lower level of physical fitness already existing prior to infection. The persistent impaired exercise capacity shown here might best be explained by multi-system dysfunction with a peripheral limitation, for example, impaired oxygen extraction due to mitochondrial dysfunction [119–121], and/or a low preceding fitness level [122], rather than a central cardiac or pulmonary limitation, but the roles of dysfunctional breathing and chronotropic incompetence need to be further investigated. In addition, it is not clear what the relatively frequent orthostatic complaints (measured via the COMPASS-31 instrument) contribute to reduced exercise capacity and how this correlates with dysfunctional breathing and chronotropic incompetence.

One of the strengths of the present study is the nested, population-based approach in defined geographic regions with a large number of individuals with PCR-confirmed earlier infection, regardless of the need for medical treatment. We focussed on adults in the working age. We avoided an overrepresentation of hospitalised elderly patients who are likely to show more SARS-CoV-2-non-specific adverse health sequelae due to more severe acute infection, comorbidities and ageing. We used within-participant comparisons considering symptom frequency before acute SARS-CoV-2 infection and considered only new symptoms not present before the acute infection. In addition, we included at least moderate severity of symptoms and considered impaired activities of daily living or work ability in our working definition of PCS. Another strength is the comprehensive clinical diagnostic work-up of both symptomatic

and symptom-free study participants, which included medical history and physical examination, laboratory investigations, CPET and a neuropsychiatric characterisation with cognitive assessment. The study allowed us to provide comparative analyses with adjustment for important confounders such as BMI, smoking and educational level and to stratify the population of persistent PCS cases by the presence of PEM (lasting >14 h) as a probably important as well as pragmatic and simple surrogate for severity.

An important limitation is that we had no objective information on exercise capacity and cognition before acute infection. We did not perform lung diffusion capacity measurements, neuroimaging or more valid measures of dysautonomia that may provide a more comprehensive understanding of the pathophysiology of PCS. Virological analyses were performed only in a subgroup and only on serum and—for a representative part of the cohort—on stool samples, but did not include the analysis of biopsy material. Furthermore, the time of sample collection >1 year post-SARS-CoV-2 infection may have precluded detection of any transient changes induced in the course of acute infection. Recall bias may be particularly relevant in individuals with more severe neurocognitive deficits. Study participation was higher by cases than by controls from phase 1, and study participants with risk factors (e.g., smoking, obesity) were less likely to respond. Another limitation is the lack of opportunities to include patients with PCS with difficulties attending the study centres because of disease severity and who would have needed admission or more support by accompanying relatives or nurses during travelling and outpatient assessment with medical tests. This might also have caused an underestimation of the prevalence of both ME/CFS and longer-lasting PEM. In addition, our screening did not include all DePaul questionnaire item scorings, which may yield PEM prevalence estimates among subjects with PCS of up to 50% or even higher [69,123–127]. We note that the selection of patients fulfilling specific PCS criteria as cases and participants with full recovery after COVID-19 and without complaints and any moderate or severe symptoms as controls (i.e., extreme phenotype selection) may lead to higher AUCs of the questionnaires when compared to representative populations. Furthermore, the population is not representative of Germany since we derived our study participants from a population of medium-sized university cities in the southwestern part of the country with substantial sociocultural and socioeconomic differences from other regions in the country. Finally, we did not include subjects from phase 1 who had symptoms compatible with PCS but did not meet the working definition criteria.

As a conclusion, we report that two-thirds of patients with PCS 6–12 months after acute SARS-CoV-2 infection continue to report persistent symptoms interfering with daily living and associated with reduced quality of life and/or work ability. The symptoms appear to change slightly but the predominant symptoms, often clustering together, remain fatigue, cognitive disturbance and chest symptoms, including breathlessness, with sleep disorder and anxiety as additional complaints in a substantial proportion of cases. In a thorough medical examination, many patients with persistent PCS show findings that significantly differ from controls and are in part abnormal/out of reference; these include impaired executive functioning, reduced cognitive processing speed and reduced physical exercise capacity only in part explained by deconditioning and typically unrelated to central cardiac or pulmonary limitations. Patients with PCS reporting PEM lasting longer than 14 h complained about more severe symptoms and showed worse findings in both cognition and exercise capacity testing. Our findings do not support hypotheses of viral persistence, EBV reactivation, adrenal insufficiency or increased complement turnover as pathophysiologically relevant for persistent PCS.

The results call for the inclusion of cognitive and exercise testing in the clinical evaluation and monitoring of patients with suspected PCS. Together with other research findings, they suggest that further studies should be undertaken to assess the role of skeletal muscle

metabolism and dysfunctional breathing as well as neurometabolic and neuroinflammatory disorders and dysautonomia for an advanced understanding of PCS development and prognosis [128,129]. Observational studies with longer follow-up are urgently needed to evaluate factors for improvement and non-recovery from PCS.

## Supporting information

**S1 STROBE Checklist. STROBE statement.**
(PDF)

**S1 Appendix. EPILOC phase 2 symptom questionnaire.**
(PDF)

**S2 Appendix. Details of clinical assessments and validated questionnaires (with references).**
(PDF)

**S3 Appendix. Methodological details of echocardiography and cardiopulmonary exercise testing (CPET) (with references).**
(PDF)

**S4 Appendix. Details of laboratory investigations.**
(PDF)

**S1 Table. Characteristics of participants with post-COVID-19 syndrome (PCS) and with recovery from phase 1 who participated in the phase 2 clinical examination.**
(PDF)

**S2 Table. Mutually adjusted predictors of case-control status change between phases 1 and 2.**
(PDF)

**S3 Table. Current medication by case-control status as reported at clinical examination in phase 2.**
(PDF)

**S4 Table. Additional results of resting heart ultrasound examination and cardiopulmonary exercise testing (CPET) analyses by case-control status as reported at clinical examination in phase 2.**
(PDF)

**S5 Table. Neurocognitive tests by cardiopulmonary exercise testing (CPET) results in participants with persistent post-COVID-19 syndrome (PCS) reported at clinical examination in phase 2.**
(PDF)

**S6 Table. Case-control status by D-dimer levels (normal vs. elevated).**
(PDF)

**S7 Table. Phase 2 case-control status by Epstein–Barr virus (EBV) and cytomegalovirus (CMV) antibody pattern.**
(PDF)

**S1 Fig. Probability of participation (95% confidence interval [CI]) by selected phase 1 characteristics.**
(PDF)

**S2 Fig. Changes in the prevalence of the five main symptom clusters (based on self-reported new symptoms of moderate-to-strong severity after acute infection) in phase 1 participants with post-COVID-19 syndrome (PCS) participating in phase 2.**
(PDF)

**S3 Fig. Individual symptoms of different grades among male and female participants with persistent post-COVID-19 syndrome (PCS), and among participants with persistent PCS with or without post-exertional malaise (PEM) lasting >14 h.**
(PDF)

**S4 Fig. Euler graphs showing the overlap of the three main symptom clusters from phase 2 based on symptoms of grade moderate-to-strong in participants with persistent post-COVID-19 syndrome (PCS) only.**
(PDF)

**S5 Fig. Euler graphs showing the descriptive prevalence (with 95% CI) of post-exertional malaise (PEM, lasting >14 h) in the various overlaps of the three main symptom clusters from phase 2 based on symptoms of grade moderate-to-strong in participants with persistent post-COVID-19 syndrome (PCS) only.**
(PDF)

**S6 Fig. Means (geometric mean for C-reactive protein [CRP]) of blood cell counts (with 95% CI) by case-control status at clinical examination in phase 2.**
(PDF)

**S7 Fig. Mean (geometric mean for glutamic-pyruvice transaminase (also known as alanine aminotranferase) [GPT], glutamic-oxaloacetic transaminase (also known as aspartate transferase) [GOT], bilirubin, dehydroepiandrosterone sulfate [DHEA-S], vitamin D and ferritin) of selected laboratory measurements (with 95% CI) by case-control status at clinical examination in phase 2.**
(PDF)

**S8 Fig. Geometric mean of anti-S1 titre (BAU/ml) by number of received vaccine doses and case-control status at clinical examination in phase 1, adjusted for sex-age class combinations, study centre and university entrance qualification.**
(PDF)

**S9 Fig. EA-D and EBNA IgG antibody levels in participants with evidence for Epstein–Barr virus (EBV) reactivation.**
(PDF)

**S10 Fig. Sex specific association of case-control status (persistent post-COVID-19 syndrome [PCS] vs. continued recovery) with abnormal neurocognitive and cardiopulmonary test results.**
(PDF)

**S1 Sensitivity Analyses. Fig A. Sensitivity analysis 1, excluding participants with health conditions already present before index infection (cardiovascular diseases, respiratory diseases, mental disorders, neurologic or sensory disorders, cancer, metabolic diseases, $n = 599$) and cases with a possible alternative medical explanation of persisting symptoms ($n = 41$). Shown are means (geometric mean for Composite Autonomic Symptom Score 31 [COMPASS-31] and Trail making test part B [TMT-B]) of self-reported health outcomes and neurocognitive tests (with 95% CI) by case-control status at clinical examination in phase 2. Fig B. Sensitivity analysis 1, excluding participants with health conditions already**

present before index infection (cardiovascular diseases, respiratory diseases, mental disorders, neurologic or sensory disorders, cancer, metabolic diseases, $n = 599$) and cases with an alternative explanation of persisting symptoms ($n = 41$). Shown are cardiopulmonary function indicators and grip strength (means with 95% CI) by case-control status at clinical examination in phase 2. Fig C. Sensitivity analysis 2, showing results for study participants with a BMI $\geq 27.5 \, \text{kg/m}^2$. Shown are means (geometric mean for COMPASS-31 and TMT-B) of self-reported health outcomes and neurocognitive tests (with 95% CI) by case-control status at clinical examination in phase 2. Fig D. Sensitivity analysis 2, results for study participants with a BMI $< 27.5 \, \text{kg/m}^2$. Shown are means (geometric mean for COMPASS-31 and TMT-B) of self-reported health outcomes and neurocognitive tests (with 95% CI) by case-control status at clinical examination in phase 2. Fig E. Sensitivity analysis 2, results for study participants with a BMI $\geq 27.5 \, \text{kg/m}^2$. Shown are cardiopulmonary function indicators and grip strength (means with 95% CI) by case-control status at clinical examination in phase 2. Fig F. Sensitivity analysis 2, results for study participants with a BMI $< 27.5 \, \text{kg/m}^2$. Shown are cardiopulmonary function indicators and grip strength (means with 95% CI) by case-control status at clinical examination in phase 2. Fig G. Sensitivity analysis 3 with persistent post-COVID-19 syndrome (PCS) additionally stratified by presence of post-exertional malaise (PEM, lasting >14 h). Shown are means (geometric mean for COMPASS-31 and TMT-B) of self-reported health outcomes and neurocognitive tests (with 95% CI) at clinical examination in phase 2. Fig H. Sensitivity analysis 3 with persistent PCS additionally stratified by presence of post-exertional malaise (PEM, lasting >14 h). Shown are cardiopulmonary function indicators and handgrip strength (means with 95% CI) at clinical examination in phase2. Fig I. Sensitivity analysis 4, in participants without medical care for their earlier acute (index) SARS-CoV-2 infection. Shown are means (geometric mean for COMPASS-31) of self-reported health outcomes (with 95% CI) by stable case-control status at clinical examination in phase 2. Fig J. Sensitivity analysis 4, in participants with medical care for their earlier acute (index) SARS-CoV-2 infection. Shown are means (geometric mean for COMPASS-31) of self-reported health outcomes (with 95% CI) by stable case-control status at clinical examination in phase 2.
(PDF)

## Acknowledgments

We thank all study participants with their care-givers and the following key collaborators (in alphabetic order) on this work: Julian Böhm, Stefan Brockmann, Stefanie Bröer, Christof Burgstahler, Katharina Caesar, Bettina Deibert, Xiaohong Du, Nelli Edel, Sabine Gerbersdorf, Jennifer Hermann, Katja Hirth, Achim Jerg, Johannes Kirsten, Manuela Licka, Jennifer Müller, Hasema Persch, Patrick Roling, Stephan Rusch, Michaela Schmid, Patrick Schneeweiß, Katarina Stete, Elisabeth Stoll, Adrian Tassoni, Hanna Tschischka, Shirin Vollrath, Vanessa Walz, Dietrich Walzer. We acknowledge the participating local laboratories and biobanking facilities for their technical support. Additional members of the EPILOC Phase 2 Study Group (in alphabetic order): Parwez Aidery (Tübingen), Daniel Bizjak (Ulm), Stefanie Bunk (Tübingen), Nadine Conzelmann (Tübingen), Stefanie Döbele (Tübingen), Lisamaria Eble (Ulm), Melanie Greibich (Heidelberg), Beate Grüner (Ulm), Lucas John (Ulm), Gerhard Kindle (Freiburg), Oliver Krumnau (Freiburg), Jessica Langel (Heidelberg), Nisar Malek (Tübingen), Moritz Munk (Ulm), Stefanie Pfau (Freiburg), Stephan Prettin (Freiburg), Hardy Richter (Tübingen), Siegbert Rieg (Freiburg), Cynthia Stapornwongkul (Freiburg), Sabine Tuma-Kellner (Heidelberg), Kay Winkert (Ulm).

## Author contributions

**Conceptualisation:** Raphael S. Peter, Alexandra Nieters, Siri Göpel, Uta Merle, Jürgen M. Steinacker, Birgit Friedmann-Bette, Claudia Schilling, Hans-Georg Kräusslich, Dietrich Rothenbacher, Winfried V. Kern.

**Data curation:** Raphael S. Peter, Alexandra Nieters, Birgit Friedmann-Bette, Barbara Müller, Claudia Schilling, Philipp Maier, Lynn Matits, Sylvia Parthé, Martin Rehm, Jana Schellenberg, Mengyu Zhu.

**Formal analysis:** Raphael S. Peter, Jürgen M. Steinacker, Barbara Müller, Sylvia Parthé, Martin Rehm, Jana Schellenberg.

**Funding acquisition:** Siri Göpel, Uta Merle, Jürgen M. Steinacker, Hans-Georg Kräusslich, Dietrich Rothenbacher, Winfried V. Kern.

**Investigation:** Siri Göpel, Uta Merle, Jürgen M. Steinacker, Peter Deibert, Birgit Friedmann-Bette, Andreas Nieß, Barbara Müller, Gunnar Erz, Roland Giesen, Veronika Götz, Karsten Keller, Philipp Maier, Lynn Matits, Sylvia Parthé, Jana Schellenberg, Ulrike Schempf, Mengyu Zhu, Hans-Georg Kräusslich, Winfried V. Kern.

**Methodology:** Raphael S. Peter, Alexandra Nieters, Jürgen M. Steinacker, Peter Deibert, Birgit Friedmann-Bette, Andreas Nieß, Barbara Müller, Claudia Schilling, Jana Schellenberg, Hans-Georg Kräusslich, Dietrich Rothenbacher, Winfried V. Kern.

**Project administration:** Alexandra Nieters, Barbara Müller, Roland Giesen, Veronika Götz, Karsten Keller, Ulrike Schempf, Hans-Georg Kräusslich.

**Resources:** Alexandra Nieters, Uta Merle, Jürgen M. Steinacker, Hans-Georg Kräusslich, Dietrich Rothenbacher, Winfried V. Kern.

**Software:** Raphael S. Peter.

**Supervision:** Siri Göpel, Uta Merle, Jürgen M Steinacker, Peter Deibert, Birgit Friedmann-Bette, Andreas Nieß, Barbara Müller, Jana Schellenberg, Hans-Georg Kräusslich, Dietrich Rothenbacher, Winfried V. Kern.

**Validation:** Raphael S. Peter, Alexandra Nieters, Siri Göpel, Jürgen M. Steinacker, Peter Deibert, Barbara Müller, Claudia Schilling, Sylvia Parthé, Jana Schellenberg, Mengyu Zhu.

**Visualisation:** Raphael S. Peter.

**Writing – original draft:** Raphael S. Peter, Alexandra Nieters, Birgit Friedmann-Bette, Barbara Müller, Claudia Schilling, Winfried V. Kern.

**Writing – review & editing:** Raphael S. Peter, Alexandra Nieters, Siri Göpel, Uta Merle, Jürgen M. Steinacker, Peter Deibert, Birgit Friedmann-Bette, Andreas Nieß, Barbara Müller, Claudia Schilling, Gunnar Erz, Roland Giesen, Veronika Götz, Karsten Keller, Philipp Maier, Lynn Matits, Sylvia Parthé, Martin Rehm, Jana Schellenberg, Ulrike Schempf, Mengyu Zhu, Hans-Georg Kräusslich, Dietrich Rothenbacher, Winfried V. Kern.

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
