## [Editor Report · Decision Letter 0]

11 Jun 2024

Dear Dr Kern, 

Thank you for submitting your manuscript entitled "Persistent symptoms and clinical findings in adults with post-acute sequelae of COVID-19/post-COVID-19 syndrome in the second year after acute infection: population-based, nested case-control study" for consideration by PLOS Medicine.

Your manuscript has now been evaluated by the PLOS Medicine editorial staff as well as by an academic editor with relevant expertise and I am writing to let you know that we would like to send your submission out for external peer review.

Please re-submit your manuscript within two working days, i.e. by Jun 13 2024 11:59PM.

Kind regards,

Katrien G. Janin, PhD

Senior Editor

PLOS Medicine

---

## [Decision Letter · Decision Letter 1]

14 Aug 2024

Dear Dr Kern,

Many thanks for submitting your manuscript "Persistent symptoms and clinical findings in adults with post-acute sequelae of COVID-19/post-COVID-19 syndrome in the second year after acute infection: population-based, nested case-control study" (PMEDICINE-D-24-01834R1) to PLOS Medicine. The paper has been reviewed by subject experts and a statistician; their comments are included below and can also be accessed here: [LINK]

After discussing the paper with the editorial team and an academic editor with relevant expertise, I'm pleased to invite you to revise the paper in response to the reviewers' comments. We plan to send the revised paper to some or all of the original reviewers (and may add an additional reviewer at a later stage), and we cannot provide any guarantees at this stage regarding publication.

We ask that you submit your revision by Sep 04 2024 11:59PM. However, if this deadline is not feasible, please contact me by email, and we can discuss a suitable alternative.

Don't hesitate to contact me directly with any questions (kjanin@plos.org). 

Best regards, 

Katrien 

Katrien Janin, PhD 

Associate Editor

PLOS Medicine

kjanin@plos.org

Comments from the academic editor:

In general, the detail of descriptive data are noteworthy just to describe the patterns being seen. 

It is difficult to make much of the associations, as many are likely confounders or due to reverse causation (as the authors point out). 

One recommendation I would make is to simply use association throughout the manuscript so readers do not conflate anything as causation (e.g., predictors doesn't seem like the best epidemiological description but is used frequently). I think also challenging to know what types of symptoms or vulnerabilities to symptoms may have existed prior to COVID even though groups are defined by reporting <80% recovery. 

It is quite interesting that there are substantial differences in the patient-reported outcomes but much less so in physiologic measurements. 

Another main recommendation would really to focus mostly on the descriptive nature of the findings (which the authors do an excellent job and provide unique ways of looking at the patterns in the data). I am not sure that these data really tell us too much about the pathophysiology of PASC (except perhaps some things that don't appear to be happening), but I think that is okay as the level of descriptive detail they are able to provide is noteworthy in and of itself.

Comments from the reviewers: 

Reviewer #1: This is a comprehensive study of the characteristics and predictors of Long COVID, and includes a large number of PROMs and objective measures. 

The introduction is relevant, methods are appropriate and the results are presented in a logical and clear manner. 

It is suitable for publication in its current format. 

It may be helpful to include the circulating variants of concern at the time this cohort was first established, as there is some evidence to suggest Long COVID risks differ across variants. 

The discussion, while relevant, is long. It may benefit from some editing down to distil the key points in a more succinct fashion. 

It would also benefit from the authors giving some hypothesis as to why educational status is associated with Long COVID outcomes. 

Nevertheless, I don't think any of the above points should necessarily be a barrier to publication. 

Reviewer #2: "Persistent symptoms and clinical findings in adults with post-acute sequelae of COVID-19/post-COVID-19 syndrome in the second year after acute infection: population-based, nested case-control study" describes a case-control study on (n=982) subjects with post-acute sequelae of COVID-19/post-COVID-19 syndrome (PCS) persisting for more than one year, in two phases. It was concluded that the majority of PCS cases did not recover in the second year of their illness. Specific strengths of the study include it prospective nature, range of data gathered, and detailed analyses. However, some issues might be considered:

1. In the Introduction section, it is stated that new symptoms were self-reported. The reliability of self-reported data might be discussed.

2. It might be clarified as to whether PCS recurrence after the study period is possible, for the controls.

3. In the Introduction section, it is stated that "We hypothesized that roughly half of the cases following the invitation would be persistent cases..."; it might be clarified as to how this estimate was obtained.

4. In the Materials and methods section, it is stated that the study categorized 28.5% of respondents as PCS, and 38% as PCS-free. The status (and clinical implications) of the remaining 33.5% might be clarified, possibly together with the definitions in the following paragraph.

5. From Supplementary figure S1, a large proportion of cases declined or were unable to attend. Might this have biased the population for Phase 2? This might be discussed.

6. Next, it is stated that "The unequal sampling ratio was based on the assumption that a significant number of phase 1 cases might have had recovered until presentation in phase 2". Again, how was this particular sampling ratio derived?

7. In the Data sources and measurements section, it is stated that vaccines received was part of the data. Was there any data on the type of vaccine?

8. For the data used, was there a possibility of any missing data? If so, how were such cases handled (exclusion, imputation)? This might be clarified.

9. In the Symptoms and signs section, symptom clusters were discussed. It might be clarified as to how these clusters were formed - were they defined beforehand, or post-hoc?

10. In the Results section, differences between cases and controls were noted. It might then be discussed as to the choice of criteria used for matching to controls, and the matching algorithm details (e.g. how much of a difference was acceptable, was multiple matching allowed, etc.)

11. In the Risk for PCS persistence subsection, "most prominent" might be "most prominently".

12. Lower educational status appears a common risk factor - might this however be due to hidden factors, e.g. type of employment?

Reviewer #3: This interesting article by Peter & Nieters and colleagues covers the clinically relevant and still timely topic of Post-COVID-Syndrome. Although the SARS-CoV-2 pandemic seems to be contained, a significant proportion of people still suffer from post-infectious symptoms. The strength of the present study is its population-based design and the detailed clinical follow-up including not only vast laboratory tests but also cardiopulmonary exercise-testing and other objective measurements. Many potential hypotheses for symptom persistence have been addressed including viral persistence, EBV reactivation, adrenal insufficiency, and increased complement turnover. The data of the current study did not show any results to support one of these hypotheses. The absence of any new evidence to further support these hypotheses maybe the most relevant news of this study. This could be highlighted in more detail in the abstract. Especially as the manuscript is rather lengthy and generally would benefit from some shortening, this essential message should not stay unrecognized for the readership. In addition, the relevance of the education level in absence of clear biomedical associations to the subjectively perceived symptoms leaves room to speculate on the pathogenesis of PCS. Based on their findings, authors maybe could include some speculations on the potential existence of different phenotypes of PCS that might be more or less related to biological vs. psycho-social aspects of each individual. 

Minor comments:

For the generalizability of the results, the flow-chart should not be part of the supplement, but be included in the main manuscript. The frequency of 3,300 cases and 4,400 controls do not sum up to the EPILOC phase 1 cohort von 11,710. Could you please elaborate on that?

---

* We ask every co-author listed on the manuscript to fill in a contributing author statement, making sure to declare all competing interests. If any of the co-authors have not filled in the statement, we will remind them to do so when the paper is revised. If all statements are not completed in a timely fashion this could hold up the re-review process. If new competing interests are declared later in the revision process, this may also hold up the submission. Should there be a problem getting one of your co-authors to fill in a statement we will be in contact. Please do not add or remove authors without first discussing this with the handling editor. You can see our competing interests policy here: http://journals.plos.org/plosmedicine/s/competing-interests .

* Please upload any figures associated with your paper as individual TIF or EPS files with 300dpi resolution at resubmission; please read our figure guidelines for more information on our requirements: http://journals.plos.org/plosmedicine/s/figures . While revising your submission, please upload your figure files to the PACE digital diagnostic tool, https://pacev2.apexcovantage.com/ . PACE helps ensure that figures meet PLOS requirements. To use PACE, you must first register as a user. Then, login and navigate to the UPLOAD tab, where you will find detailed instructions on how to use the tool. If you encounter any issues or have any questions when using PACE, please email us at PLOSMedicine@plos.org.

* Please ensure that the paper adheres to the PLOS Data Availability Policy (see http://journals.plos.org/plosmedicine/s/data-availability ), which requires that all data underlying the study's findings be provided in a repository or as Supporting Information. For data residing with a third party, authors are required to provide instructions with contact information (web or email address) for obtaining the data. Please note that a study author cannot be the contact person for the data. PLOS journals do not allow statements supported by "data not shown" or "unpublished results." For such statements, authors must provide supporting data or cite public sources that include it.

* At this stage, we ask that you include a short, non-technical Author Summary of your research to make findings accessible to a wide audience that includes both scientists and non-scientists. The Author Summary should immediately follow the Abstract in your revised manuscript. This text is subject to editorial change and should be distinct from the scientific abstract. Ideally each sub-heading should contain 2-3 single sentence, concise bullet points containing the most salient points from your study. In the final bullet point of 'What Do These Findings Mean?', please include the main limitations of the study in non-technical language. Please see our author guidelines for more information: https://journals.plos.org/plosmedicine/s/revising-your-manuscript#loc-author-summary .

FIGURES AND TABLES

SUPPLEMENTARY MATERIAL

REFERENCES

* Please ensure that journal name abbreviations match those found in the National Center for Biotechnology Information (NCBI) databases (http://www.ncbi.nlm.nih.gov/nlmcatalog/journals ), and are appropriately formatted and capitalised.

OBSERVATIONAL STUDIES

* Abstract: Please include the study design, population and setting, number of participants, years during which the study took place (enrollment and follow up), length of follow up, and main outcome measures.

* Please ensure that the study is reported according to the STROBE (or appropriate STOBE extension) guideline (available from: https://www.equator-network.org/reporting-guidelines/strobe ) and include the completed STROBE (or STROBE extension) checklist as Supporting Information. Please add the following statement, or similar, to the Methods: "This study is reported as per the Strengthening the Reporting of Observational Studies in Epidemiology (STROBE) guideline (S1 Checklist)." When completing the checklist, please use section and paragraph numbers, rather than page numbers. 

* For all observational studies, in the manuscript text, please indicate: (1) the specific hypotheses you intended to test, (2) the analytical methods by which you planned to test them, (3) the analyses you actually performed, and (4) when reported analyses differ from those that were planned, transparent explanations for differences that affect the reliability of the study's results. If a reported analysis was performed based on an interesting but unanticipated pattern in the data, please be clear that the analysis was data driven. 

* Please state in the Methods section whether the study had a prospective protocol or analysis plan. If a prospective analysis plan (from your funding proposal, IRB or other ethics committee submission, study protocol, or other planning document written before analyzing the data) was used in designing the study, please include the relevant document(s) with your revised manuscript as a Supporting Information file to be published alongside your study and cite it in the Methods section. A legend for this file should be included at the end of your manuscript. If no such document exists, please make sure that the Methods section transparently describes when analyses were planned, and when/why any data-driven changes to analyses took place. Changes in the analysis, including those made in response to peer review comments, should be identified as such in the Methods section of the paper, with rationale.

---

## [Decision Letter · Decision Letter 2]

26 Sep 2024

Dear Dr. Kern,

Thank you very much for submitting your revised manuscript "Persistent symptoms and clinical findings in adults with post-acute sequelae of COVID-19/post-COVID-19 syndrome in the second year after acute infection: population-based, nested case-control study" (PMEDICINE-D-24-01834R2) to PLOS Medicine.

I have discussed the paper with my colleagues and the academic editor and it was also seen again by two of the original reviewers. I am pleased to inform you that we plan to accept the paper for publication in the journal, provided the remaining editorial and production issues are resolved. The remaining issues that need to be addressed are listed at the end of this email. Any accompanying reviewer attachments can be seen via the link below. 

[LINK]

When you submit the revised manuscript, please provide a point-by-point response to the reviewers’ and editors’ comments, indicating the changes you have made in the manuscript. Please submit a clean version of the paper as the main article file, and a tracked version as a marked-up manuscript file.

We ask that you submit your revised manuscript within 1 week (October 4th). Please me directly at hvanepps@plos.org if you have any questions or concerns or you need to request an extension to the deadline. We look forward to receiving your revised manuscript.

Kind regards,

Heather

Heather Van Epps, PhD

Executive Editor 

PLOS Medicine

hvanepps@plos.org

Comments from Reviewers:

Reviewer #2: 

We thank the authors for addressing our previous concerns. Some additional comments follow:

1. More specific details on the hypothesis that half of the cases would be persistent might be stated, in particular the literature consulted, and any study-specific factors considered.

2. It is clarified that missing observations were excluded in specific analyses. Where possible, the proportion of missing data for each relevant variable might be stated.

3. From Supplementary table S2, two or more vaccine shots appears correlated with a lower odds ratio for cases improved, and a higher odds ratio for controls worsening (although the effect may not be significant). This observation might be briefly discussed if appropriate.

Reviewer #3: 

I have no further comments.

[LINK] 

Requests from Editors:

1. Data Availability Statement: Please provide instructions with contact information (web or email address) for obtaining the data. Please note that a study author cannot be the contact person for the data. 

2. General: please ensure that you use patient centric language thoughout the paper. For example, you should avoid referring to individuals as ‘cases’ and ‘controls’ (throughout paper; eg, in the introduction “PCS patients” should be “patients with PCS”).

3. Page 16, “...there was a statistically significant association for CRP, HbA1c, and D-dimers before adjustment for BMI and smoking (data not shown).” PLOS journals do not allow statements supported by "data not shown" or "unpublished results." For such statements, authors must provide supporting data (in the supplement) or cite public sources that include it. 

4. Abstract: Please combine the Methods and Findings sections into one section, per PLOS Medicine style.

5. Abstract: Please spell out ME/CFS (also at first mention in the main text).

6. Abstract: Please consider adding ‘relative to controls’ to the end of the following sentence, “In persistent cases, handgrip strength, maximal oxygen consumption, and ventilator efficiency were significantly reduced”. 

7. Abstract: Please spell out post-exertional malaise throughout, in order to minimize the use of abbreviations.

8. Please revise the Author summary so that it conforms to PLOS Medicine style (headings and bullet points). The Author Summary should immediately follow the Abstract in your revised manuscript. This text is subject to editorial change and should be distinct from the scientific abstract. Ideally each sub-heading should contain 2-3 single sentence, concise bullet points containing the most salient points from your study. In the final bullet point of ‘What Do These Findings Mean?’, please include the main limitations of the study in non-technical language. Please see our author guidelines for more information: https://journals.plos.org/plosmedicine/s/revising-your-manuscript#loc-author-summary

9. Results, p 10: Please insert the exact value where you state “circa 65%”

10. Results, p 10, line 5: Please remove ‘respectively’ at the end of the sentence, as it is not necessary

11. Results, p 13: It is unclear what you mean by “somehow overrated” and “somehow underrated” with regard to specific symptoms comparing self-report and validated questionnaires. Please consider revising the wording in the interest of clarity. 

12. The Supplementary text does not appear to have been carried over from the previous revision; the supplementary file includes only display items (tables, figures). Please provide the supplement as one continuous PDF file. 

13. Acknowledgments section: Does the list provided in this section include patients? If so, do you have permission to disclose their names?

14. Please ensure that the study is reported according to the STROBE (or appropriate STOBE extension) guideline (available from: https://www.equator-network.org/reporting-guidelines/strobe ) and include the completed STROBE (or STROBE extension) checklist as Supporting Information. When completing the checklist, please use section and paragraph numbers, rather than page numbers. Please add the following statement, or similar, to the Methods: "This study is reported as per the Strengthening the Reporting of Observational Studies in Epidemiology (STROBE) guideline (S1 Checklist)." When completing the checklist, please use section and paragraph numbers, rather than page numbers

---

## [Editor Report · Decision Letter 3]

24 Oct 2024

Dear Dr. Kern,

Thank you very much for re-submitting your manuscript "Persistent symptoms and clinical findings in adults with post-acute sequelae of COVID-19/post-COVID-19 syndrome in the second year after acute infection: population-based, nested case-control study" (PMEDICINE-D-24-01834R3) for review by PLOS Medicine.

I am pleased to say that provided the remaining editorial and production issues are dealt with in full, we are planning to accept the paper for publication in the journal.

[LINK]

Please ensure that the paper adheres to the PLOS Data Availability Policy (see http://journals.plos.org/plosmedicine/s/data-availability ), which requires that all data underlying the study's findings be provided in a repository or as Supporting Information. For data residing with a third party, authors are required to provide instructions with contact information for obtaining the data. PLOS journals do not allow statements supported by "data not shown" or "unpublished results." For such statements, authors must provide supporting data or cite public sources that include it.

If you have any questions in the meantime, please contact me (lgaynor@plos.org) or the journal staff on plosmedicine@plos.org.  

We look forward to receiving the revised manuscript by Oct 31 2024 11:59PM.   

Sincerely,

Louise Gaynor-Brook, MBBS PhD

Senior Editor 

PLOS Medicine

plosmedicine.org

Requests from Editors:

Please address the first two comments from Reviewer 2 by incorporating the suggested changes. We note that Nehme et al (Ann Intern Med 2021) and Haverall et al (JAMA 2021) were major sources considered when planning your study, yet are not cited in your reference list. A detailed explanation is not required; it would be satisfactory to include that hypotheses were based on the available literature (with citations) and from your clinical experience at the centres mentioned. In addition, it would be helpful to include % missing data in the relevant tables, as per the reviewer’s suggestion. 

Please see below a list of further minor points requiring attention. The list below may appear lengthy, but most of these are more minor points which should not require a substantial amount of time to attend to.

General comments:

Please replace "subject" with participant, patient, individual, or person throughout the manuscript.

Please use person-first language throughout e.g. patients with persistent PCS. 

Throughout the paper, please adapt reference call-outs to the following style: "... every year [1,2]." (noting the absence of spaces within the square brackets, and placement of the reference call-out preceding punctuation).

Please include continuous line numbers in your revised manuscript. 

To help us extend the reach of your research, please provide any Twitter handle(s) that would be appropriate to tag, including your own, your co-authors’, your institution, funder, or lab.

Title: Please revise your title to: “Persistent symptoms and clinical findings in adults with post-acute sequelae of COVID-19/post-COVID-19 syndrome in the second year after acute infection: A population-based, nested case-control study”

Abstract:

Please note that the word limit for Abstracts (500 words) can be slightly exceeded to allow the following changes to be incorporated. We suggest also presenting the most salient findings within the Abstract. 

Please structure your abstract using the PLOS Medicine headings (Background, Methods and Findings, Conclusions).

Abstract Background: Please provide some brief context of why the study is important. 

Please omit 'were' from "New symptoms or PCS development among control subjects was associated with were an intercurrent..."

Please define all abbreviations at first use. 

Please ensure that all numbers presented in the abstract are present and identical to numbers presented in the main manuscript text.

Please provide brief demographic details of the study population (e.g. sex, age, ethnicity, etc)

Please quantify the main results (with 95% CIs and p values).

Please provide the range for length of time between phases 1 and 2, in addition to the median.

Please include the important dependent variables that are adjusted for in the analyses.

In the last sentence of the Abstract Methods and Findings section, please describe 2-3 of the main limitations of the study's methodology.

Please begin your Abstract Conclusions with "In this study, we observed ..." or similar, to summarize the main findings from your study, without overstating your conclusions. Please emphasize what is new and address the implications of your study, being careful to avoid assertions of primacy.

Author Summary:

Thank you for providing an Author Summary. 

Please temper assertions of primacy by adding ‘to the best of our knowledge’ or similar to the sentence “The long-term prognosis of this post-COVID-19 syndrome (PCS) is unknown.” 

Please revise “...adjustment for possible confounders….” to include a couple of examples. 

Please revise to “...but we did not identify major pathology in laboratory investigations.”

Methods:

Did your study have a prospective protocol or analysis plan? Please state this - either way - early in the Methods section. If a prospective analysis plan (from your funding proposal, IRB or other ethics committee submission, study protocol, or other planning document written before analyzing the data) was used in designing the study, please include the relevant prospectively written document with your revised manuscript as a Supporting Information file to be published alongside your study, and cite it in the Methods section. A legend for this file should be included at the end of your manuscript. If no such document exists, please make sure that the Methods section transparently describes when analyses were planned, and if/when reported analyses differed from those that were planned. Changes in the analysis-- including those made in response to peer review comments-- should be identified as such in the Methods section of the paper, with rationale. If a reported analysis was performed based on an interesting but unanticipated pattern in the data, please be clear that the analysis was data-driven.

Please provide a copy of the original symptom questionnaire as a supplementary file. We do not require copies of validated questionnaires. 

Please define ACTH and DHEA-S at first use.

Please add the following statement, or similar, to the Methods: "This study is reported as per the Strengthening the Reporting of Observational Studies in Epidemiology (STROBE) guideline (S1 Checklist)." 

Results: 

Page 11 - Please revise to “...factors associated with improvement (either to intermediate or control status)...”.

Page 12 - Please revise to “...or as adjusted average ratings (figure 3).”

Page 12 - Please remove ‘somehow’ from “...symptom cluster varied somehow…“.

Page 14 - please avoid use of causative language; please revise to “...including the perceived impact of any illnesses or adverse condition …” 

Page 14 - Please revise to “...grip strength was lower in women than men (30.9kg versus 50.7 kg)...”.

Page 14 - Please revise “...(adjusted systolic mean: 80 to 81 mmHg for all subgroups, p=0.88; diastolic mean: 128 to 131 mmHg for all subgroups, p=0.21).“ - please clarify values for SBP and DBP.

Page 14 - For paragraph beginning “Differences were observed for FEV1 and FVC, SpO2 at rest, and several CPET derived variables”, please clarify in which table results are provided, as this does not appear to be Table S4 as currently stated. Please quantify results with p values and 95% CIs, at least in the tables.

Discussion:

Please present and organize the Discussion as follows: a short, clear summary of the article's findings; what the study adds to existing research and where and why the results may differ from previous research; strengths and limitations of the study; implications and next steps for research, clinical practice, and/or public policy; one-paragraph conclusion.

Please remove all subheadings within your Discussion e.g. Trajectories.

Please remove the information on competing interests and funding from the end of the main text. In the event of publication, this information will appear in the article metadata, via entries in the submission form.

Page 19 - please revise ‘vaccine shots ‘ to ‘vaccine doses’.

Figures:

Please consider avoiding the use of red and green in order to make your figure more accessible to those with colour blindness.

Please define all abbreviations used in the figure legend of each figure.

Please clarify what is represented by the numbers within brackets in Figures 3 and S5, and add to the figure legend. 

Please indicate in the figure caption the meaning of the bars in Figure S6 and S8. 

Tables:

Please define all abbreviations used in the table legend of each table.

Please provide the unadjusted comparisons as well as the adjusted comparisons in Tables S2, S4.

Where adjusted analyses are presented, please specify the variables controlled for in the table legend. 

Please clarify what is represented by the numbers within brackets in the tables e.g. %.

When a p value is given, please specify the statistical test used to determine it in the figure legend.

References:

Please ensure that journal name abbreviations match those found in the National Center for Biotechnology Information (NCBI) databases (http://www.ncbi.nlm.nih.gov/nlmcatalog/journals ), and are appropriately formatted and capitalised. The first six authors should be listed prior to ‘et al’. 

Please also see https://journals.plos.org/plosmedicine/s/submission-guidelines#loc-references for further details on reference formatting. 

Supplementary files: 

Please note that supplementary figures S10 to S20 are not provided, but are referred to in the main text. Use whole numbers when naming your supporting information files. Combine separate parts (e.g., S11A and S11B) into one file (e.g. S11 Figure) or rename with whole numbers (e.g., Figure S11, Figure S12, and so on).

[LINK]

---

## [Editor Report · Decision Letter 4]

17 Dec 2024

Dear Dr Kern, 

On behalf of my colleagues and the Academic Editor, Aaloke Mody, I am pleased to inform you that we have agreed to publish your manuscript "Persistent symptoms and clinical findings in adults with post-acute sequelae of COVID-19/post-COVID-19 syndrome in the second year after acute infection: a population-based, nested case-control study" (PMEDICINE-D-24-01834R4) in PLOS Medicine.

PRESS

We also ask that you take this opportunity to read our Embargo Policy regarding the discussion, promotion and media coverage of work that is yet to be published by PLOS. As your manuscript is not yet published, it is bound by the conditions of our Embargo Policy. Please be aware that this policy is in place both to ensure that any press coverage of your article is fully substantiated and to provide a direct link between such coverage and the published work. For full details of our Embargo Policy, please visit http://www.plos.org/about/media-inquiries/embargo-policy/ .

Sincerely, 

Rebecca Kirk

On behalf of:

Louise Gaynor-Brook, MBBS PhD 

Senior Editor 

PLOS Medicine